# Basal ganglia output (entopeduncular nucleus) coding of contextual kinematics and reward in the freely moving mouse

Anil K Verma Rodriguez[1], Josue O Ramírez-Jarquin[1], Román Rossi-Pool[2], Fatuel Tecuapetla[1]*

[1]Instituto de Fisiología Celular, Departamento de Neuropatología Molecular, Universidad Nacional Autónoma de México, Mexico city, Mexico; [2]Instituto de Fisiología Celular, Departamento de Neurociencia Cognitiva, Universidad Nacional Autónoma de México, Mexico City, Mexico

## eLife Assessment

This **valuable** study reports on electrophysiological recording of the spiking activity of single neurons in the entopeduncular nucleus (EPN) in freely-moving mice performing an auditory discrimination task. The data show that the activity of single EPN neurons is modulated by reward and movement kinematics, with the latter further affected by task contexts (e.g. movement toward or away from a reward location). The results provide **solid** evidence for the conclusions. There is some ambiguity as to whether the data contain the population of EPN neurons characterized in previous studies that obtained different results. Investigations separating confounding factors would be of benefit. Nonetheless, the work is overall of interest to those who study how the basal ganglia, particularly the EPN, contribute to behavior.

*For correspondence:
fatuel@ifc.unam.mx

Competing interest: The authors declare that no competing interests exist.

**Abstract** The entopeduncular nucleus (EPN) is often termed as one of the output nuclei of the basal ganglia owing to their highly convergent anatomy. The rodent EPN has been implicated in reward and value coding whereas the primate analog internal Globus Pallidus has been found to be modulated by some movements and in some circumstances. In this study, we sought to understand how the rodent EPN might be coding kinematic, reward, and difficulty parameters, particularly during locomotion. Furthermore, we aimed to understand the level of movement representation: whole-body or specific body parts. To this end, mice were trained in a freely moving two-alternative forced choice task with two periods of displacement (return and go trajectories) and performed electrophysiological recordings together with video-based tracking. We found (1) robust reward coding but not difficulty. (2) Spatio-temporal variables better explain EPN activity during movement compared to kinematic variables, while both types of variables were more robustly represented in reward-related movement. (3) Reward-sensitive units encode kinematics similarly to reward-insensitive ones. (4) Population dynamics that best account for differences between these two periods of movement can be explained by allocentric references like distance to reward port. (5) The representation of paw and licks is not mutually exclusive, discarding a somatotopic muscle-level representation of movement in the EPN. Our data suggest that EPN activity represents movements and reward in a complex way: highly multiplexed, influenced by the objective of the displacement, where trajectories that lead to reward better represent spatial and kinematic variables. Interestingly, there are intertwining representations of whole-body movement kinematics with a single paw and licking variables. Further, reward-sensitive units encode kinematics similarly to reward-insensitive ones, challenging the notion of distinct pathways for reward and movement processing.

## Introduction

The EPN is often termed as the output of the basal ganglia. The basal ganglia are a group of subcortial nuclei with highly converging anatomy that have been associated with motor and non-motor functions. Whereas the striatum receives inputs from many diverse brain regions (*Hintiryan et al., 2016*; *Hunnicutt et al., 2016*) most of its projections end in the internal Globus Pallidus (GPi, primate analog of EPN) or substantia nigra pars reticulata (SNr, often termed as the output nuclei) through the direct and indirect pathways. This is accompanied by a sharp reduction in the number of neurons: the striatum has 2–3 orders of magnitude more neurons. This highly convergent anatomy suggests information synthesis/convergence.

Previous studies focusing on the non-motor coding of the internal Globus Pallidus, have found that there is coding of reward (*Hong and Hikosaka, 2008*). Rodent studies have corroborated this finding and suggested the existence of value coding that serves to evaluate actions (*Stephenson-Jones et al., 2016*).

There is a particular interest in understanding the role of the basal ganglia on movement production since several diseases with motor disturbances have been linked to basal ganglia malfunction. Several studies have found a relationship between the GPi activity and limb movements of the primate. However, studies have found inconsistencies in the coding of movement depending on movement type (ramping vs. stepped, self-paced vs step tracking, *Georgopoulos et al., 1983*) or between identical movements performed under different cognitive states (cued vs memory-dependent *Turner and Anderson, 2005*). Further, some studies agree that self-paced movements are represented in a weaker manner (*Mink and Thach, 1991*; *Turner and Anderson, 2005*). Overall, findings show that GPi units are selectively engaged in some types of movement, but it is so far unclear what cognitive contingency is most engaging.

On the other hand, current theories of the overall function of the basal ganglia, like the 'rate theory' (*Albin et al., 1989*) or the 'dynamic activity model' (*Nambu et al., 2023*), posit that the firing rate of the output nuclei is inversely related to movement facilitation.

In this study, we sought to examine how the information regarding kinematics, particularly during locomotion, was represented in the entopeduncular nucleus of freely moving mice. Given that previous studies have highlighted the importance of reward-related activity, we further sought to understand whether reward-sensitive units would be insensitive to kinematic coding. Finally, we investigated whether the EPN units were modulated by individual movements (limb or licking) as opposed to whole-body movements.

## Results

Given that our main interest lies in how the EPN neuronal activity encodes kinematic and reward-related variables we set up a task and recorded activity in freely moving mice. We first describe the population dynamics that underlie reward, followed by asking how movement is encoded by the units, contrasting both periods of movement in the designed task (Return and Go trajectories). Next, we focus on extracting the population dynamics that make Return and Go trajectories different. Finally, we describe how EPN units are modulated by individual movements by analyzing gait and licking behavior.

### Mice can perform psychophysical responses on a self-paced two-alternative forced choice task based on frequency-modulated sweeps

The main goal of this study was to establish the relationships between kinematic variables of mice, particularly during locomotion, and the activity of the entopeduncular nucleus. However, given that previous studies have ascertained that reward is an important variable coded in this nucleus, we sought to evaluate how reward might be related to kinematic coding.

To this end, we devised a task in which animals were required to displace towards different corners of a triangular arena (side = 35 cm, *Figure 1A–B*). In a trial, animals were required to move to the Waiting Corner and wait for ~1 s (0.8–1.5 s). After waiting, an auditory stimulus would play, which would indicate in which other corner a water reward would be available (a two-alternative forced choice). If the animal arrived at the appropriate lick port, it was termed a correct response, and a water droplet (3 µl) was then released. Incorrect responses were signaled by a 10–20 s timeout during which

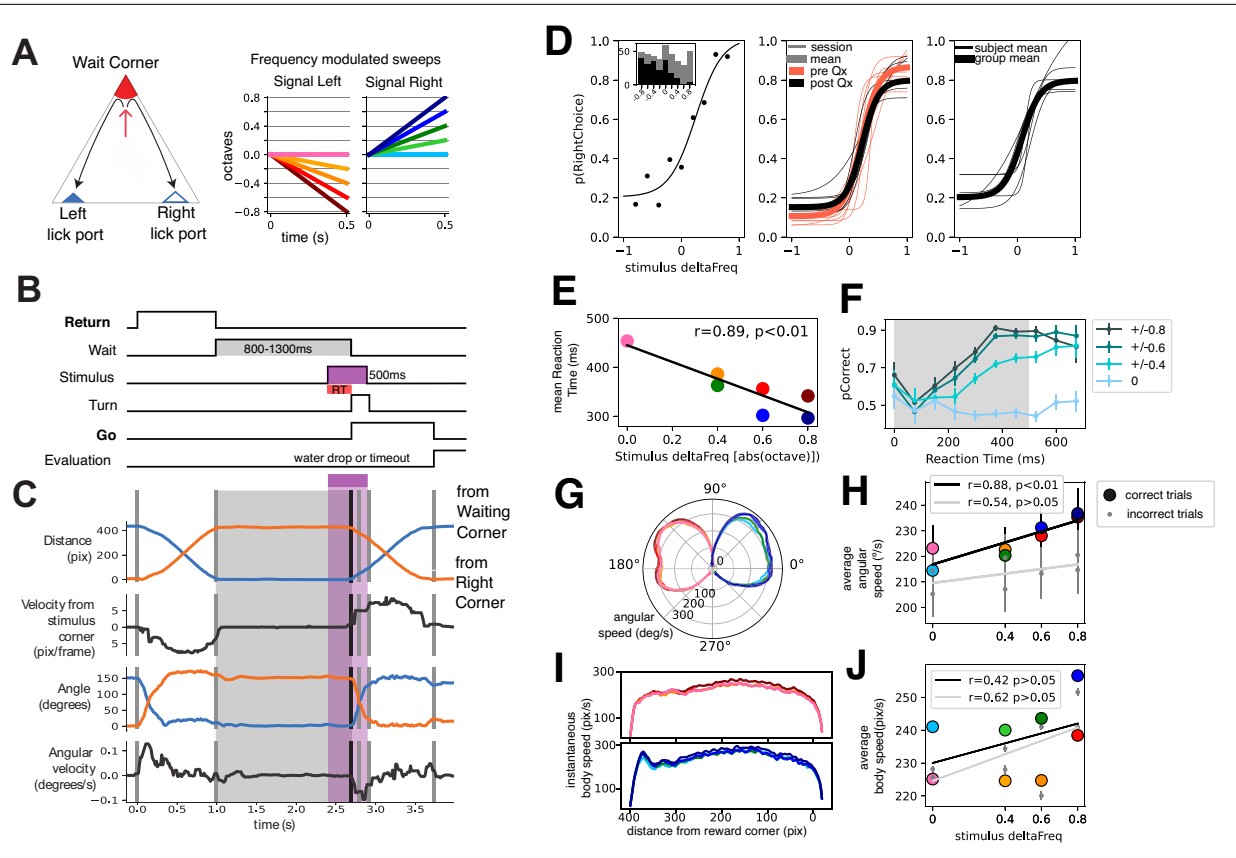

**Figure 1.** Mice learn to perform a two-alternative forced choice task in a freely moving triangular arena. (**A**) Left: Schematic of triangular arena. Right: stimuli were frequency-modulated sweeps. (**B**) Time schematic of task events. The two periods of movement are highlighted. (**C**) Video tracking extracted variables of a single trial. Note that events correspond to B. (**D**) Left: Psychophysical performance of an example session of an animal. Middle: Thin traces are five example days pre (orange) and post (black) electrode implant for an example animal. Thicker traces are averages of pre and post-performance. Right: Thin traces are individual animal averages (n=6) of the 32 recording sessions included in this study. Thicker trace is group average. (**E**) Average reaction time as a function of stimulus deltaFreq. Black line is the linear fit. (**F**) Probability of correct response as a function of reaction time. Traces are grouped per absolute deltaFreq (left and right responses are grouped). Probability of correct responses is computed in reaction time bins of 75 ms, binned reaction time from n=38049 trials, from n=6 animals. (**G**) Instantaneous angular speed (radial axis) of turning per stimulus condition. 90° represents the animals' position during waiting. (**H**) Mean angular speed per stimulus identity for correct (colored circles) and incorrect (gray circles) trials. Trendlines were drawn following a simple linear regression. Legend shows correlation coefficient and statistical significance, binned angular velocity from n=38049 trials, from n=6 animals. (**I**) Instantaneous displacement speed across the length of the trajectory for left and right stimuli. (**J**) Mean average speed per stimulus for correct (colored circles) and incorrect (gray circles) trials, binned average body speed from n=38049 trials, from n=6 animals. Trendlines were drawn following a simple linear regression. Legend shows correlation coefficient and statistical significance.

ambient lights were dimmed. Animals would then start another trial (by moving to the Waiting Corner) in a self-paced manner. By performing tracking DeepLabCut, (*Mathis et al., 2018*) of video recordings from a bottom-up view (see *Video 1*), we could measure several variables for each trial (distance to different corners, angular velocity, *Figure 1C*) which allowed to establish different events of the task (*Figure 1B*). Particularly, we could establish two distinct periods of movement/displacement: Return trajectories (to reach the wait corner) and Go trajectories (to reach a reward corner).

Auditory stimuli consisted of 0.5 s frequency-modulated sweeps; upward and downward frequency modulation indicated a right and left choice, respectively (*Figure 1A*, right). By varying the rate at which the frequency was modulated we attempted to modulate the certainty with which animals decided. This was evidenced by a psychometric curve fitted to the response probability as a function of stimulus deltaFreq (*Figure 1D*). Note that the central stimulus was a pure tone that had no information of side. This behavior was reproducible for different sessions before and after electrode implant (*Figure 1D*, middle) and in several animals included in this study (n=6; *Figure 1D*, right).

We then quantified how the different stimuli influenced response time and kinematic parameters. Note that stimulus pairs with higher deltaFreq (+/-0.8) were easier than those with less (+/-0.4)

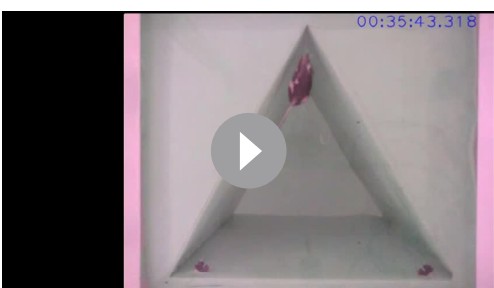

**Video 1.** Example trials of the task. Freely moving two-alternative forced choice task with two periods of displacement. Animals were required to displace towards different corners of a triangular arena. In a trial, animals were required to move to the Waiting Corner and wait for ~1 s (0.8–1.5 s). After waiting, an auditory stimulus would play, which would indicate in which other corner a water reward would be available. If the animal arrived at the appropriate lick port, it was termed a correct response, and a water droplet was then released. Incorrect responses were signaled by a 10–20 s timeout during which ambient lights were dimmed. Animals would then start another trial (by moving to the Waiting Corner) in a self-paced manner. https://elifesciences.org/articles/98159/figures#video1

or no deltaFreq. The +/-0.2 stimulus pair was dropped from some sessions to increase trials per condition and will not be further analyzed. We quantified Response time as the time from stimulus onset to the onset of turning (*Figure 1B–C*), which we defined as the moment when animals had committed to a motor action (either left or right turn). We found a relationship between response time and the stimulus difficulty (correlation coefficient *r*=0.89, p<0.01 permutation test; *Figure 1E*). To ascertain whether the evolving deltaFreq of the stimuli had an impact on performance, we calculated the probability of a correct response as a function of response time by binning trials according to response time (bin = 75 ms) and by stimulus difficulty (deltaFreq). For the central stimulus, no amount of response time increases the probability of correct response since the stimulus contains no information on the correct side. For stimuli pairs with varying difficulty, there is a graded increase in the probability of a correct response (*Figure 1F*).

Next, we sought to ask whether the difficulty of the stimuli and thus uncertainty in the decision could modulate the kinematic response of the animals. We hypothesized that responses to easier stimuli would result in faster movement. We calculated the instantaneous angular velocity across the 150° of turning that animals had to perform (*Figure 1G*), showing indeed a subtle gradation according to the difficulty of the stimuli. We found a significant correlation between average angular speed and the deltaFreq of stimuli for correct trials but not so for incorrect trials (*Figure 1H*, black and gray lines, respectively). We further calculated the instantaneous speed across the trajectory that animals had to complete (Go trajectory, *Figure 1I*). Animals performed easy trials faster than more difficult ones, but no significant correlation between stimulus difficulty and speed was found (*Figure 1J*).

From the performance of animals in this task we conclude that animals can perform a psychophysical task based on frequency-modulated sweeps, with easier stimuli resulting in greater hits than more difficult stimuli. The difficulty of the stimuli had an impact on the reaction time, and increasing the sampling time for stimuli with information of correct response increased the probability of performing a correct choice. Furthermore, difficulty had a moderate impact on the way animals turned, but not on the whole period of locomotion.

## EPN recordings and examples

After a consistent performance across at least 2 wk (*Figure 1D*) animals were implanted with a movable microwire bundle to record the activity of EPN neurons. The microwires were advanced in 50 micrometer steps for 10 steps in total. We inferred the recording position via postmortem histological analysis of the individual microwire tracks and included for further analysis only the ones that were recorded inside of EPN (*Figure 2B*). Previous reports indicated that the two main cell types in the EPN characterized by parvalbumin + and somatostatin + markers, (*Stephenson-Jones et al., 2016*; *Wallace et al., 2017*) exhibit differences in the coefficient of variation and the duration of the action potentials. To this end, we computed these characteristics for the units recorded in this study (*Figure 2C*). However, we could not separate the population into two distinct clusters.

*Figure 2D–E* shows two examples of the activity recorded aligned to the task-relevant events identified in *Figure 1B*: return, wait, turn (0°, 75°, 150°), outcome. Note that we sought to isolate the turn during the Go trajectory by identifying the start (0°), middle (75°), and end (150°) of the turn; the top row shows the raster plots color-coded by the stimulus for all correct trials. Despite the performance of animals being highly stereotyped, there are slight differences due to it being freely moving. Thus,

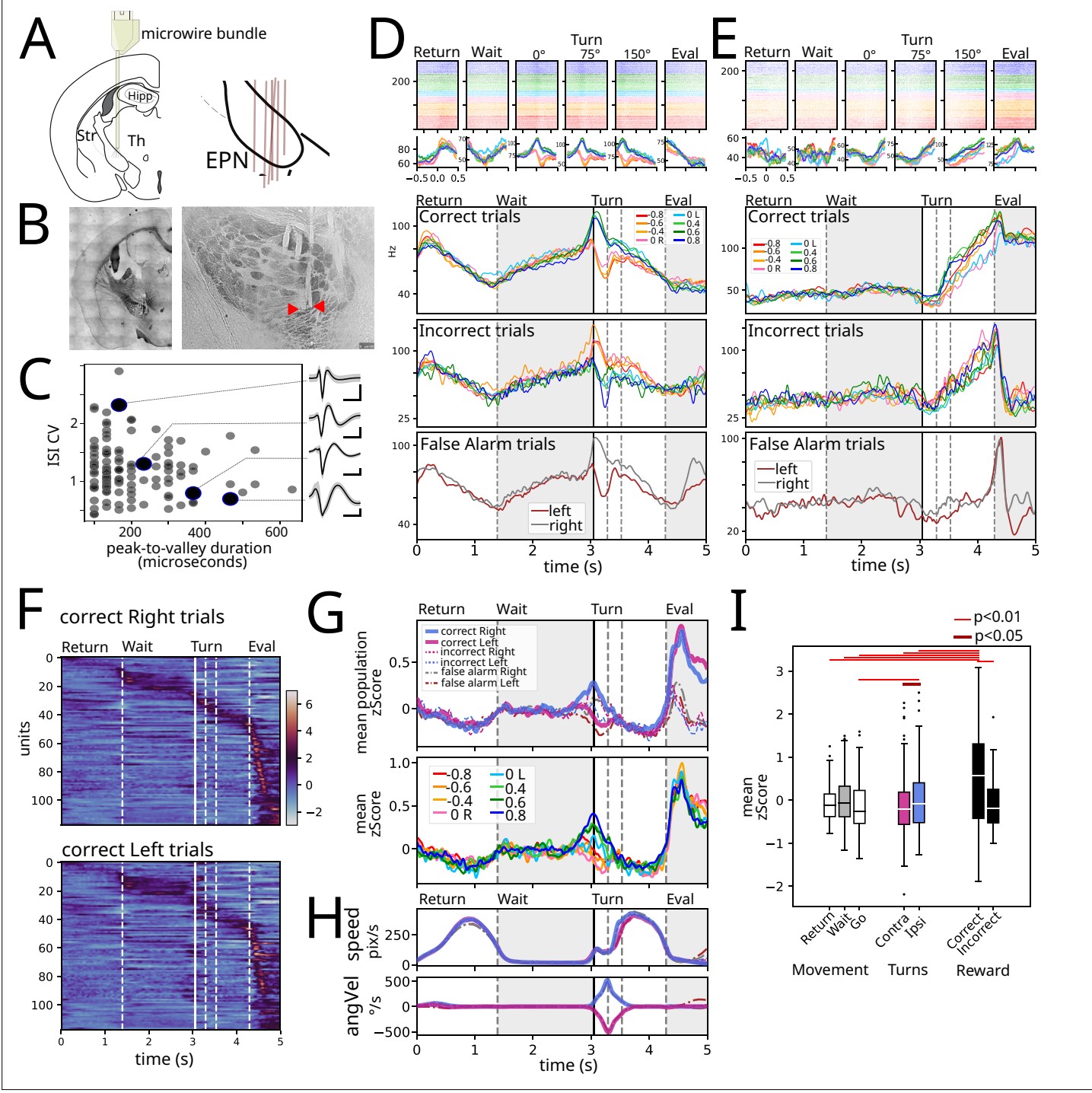

**Figure 2.** Electrophysiological recordings were obtained from the entopeduncular nucleus (EPN). (**A**) Microwire bundles were implanted onto the entopeduncular nucleus. Right pane shows a schematic of electrode placement (n=6). (**B**) Left: Photomicrograph of electrode cannula track. Right: Close-up of individual microwire electrode tracks. Red arrow heads point to microwire tips. (**C**) Peak to valley duration of plotted against inter-spike interval coefficient of variation for recorded spikes. Insets: average waveform for four example units. (**D**) Example recorded unit aligned to events identified in *Figure 1B*: return, wait, turn (0°, 75°, 150°), evaluation, averaged according to stimulus presented. First row is rescaled average for correct trials. Second row is average of incorrect trials. Third row is false alarm trials: trials where animals did not wait long enough for the stimulus to appear but that performed the entire movement sequence either to the right or left lick ports. (**E**) Same as D for another example unit. (**F**) Population response for correct left and right responses. (**G**) Upper: Average z-score segregated by left and right, for correct, incorrect, and false alarm trials. Lower: Average z-score per stimulus for correct trials. (**H**) Average speed (upper) and angular velocity (lower) for trials segregated as in the upper pane of G. (**I**) Mean population activity per epoch segregated by movement, turning, and Reward. Wilcoxon signed-rank test used, n = 118 units per condition.

in the three bottom rows, we show the rescaled activity presented in the raster plots for the correct trials, and additionally for the incorrect trials and false alarm trials, respectively. The false alarm trials were trials in which animals did not wait for the auditory stimulus to be presented and performed the entire motor sequence identical to real trials (useful since they are gated by an internal signal without external stimuli with identical motor output).

The unit in *Figure 2D* shows a decreasing firing rate as time/distance from the goal of each of the Return (reach the wait corner) and Go trajectories (reach the reward corner). Further, this unit shows differential activity during left and right turns that follows the direction of turning in correct, incorrect, and false alarm trials. On the other hand, the unit in *Figure 2E* exhibited a marked difference in the time/distance varying activity during Return and Go trajectories (with no change and ramping activity, respectively). Further, this second unit exhibits a stark difference in the evaluation period between rewarded and unrewarded trials.

Heatmaps in *Figure 2F* show responses for all recorded units (118 units from 6 animals) averaged for all correct right (*Figure 2F*, upper) and left (*Figure 2F*, lower) trials, sorted by peak activity in rightward trials. Even if neurons are organized employing only the rightward trials, the population activity displays akin responses. *Figure 2G* upper panel shows averaged z-score EPN activity sorted by turn side and outcome (correct, incorrect, and false alarms), with average kinematic measurements of speed and angular velocity presented in *Figure 2H*. These population averages show that there is population coding associated to turn and reward. Sorting hit trials based on stimulus identity (*Figure 2G*, lower) preserves side representation; however, differences across stimuli are remarkably low.

We sought to assess how average population activity could be representing movement, turning, and reward. Thus, we obtained the z-score population average during both periods of movement (Return and Go) and the wait period (*Figure 2I*). As well, we obtained the average activity during turns ipsilateral and contralateral to the recording site (right hemisphere), and for correct and incorrect trials during the evaluation period. Out of all conditions, reward (evaluation of correct trials) exhibits the highest population activity (p<0.01, Wilcoxon test); correct trials also exhibit a higher activity compared to unrewarded trials despite similar speed (*Figure 2H*). Ipsilateral turns exhibit a higher activity compared to contralateral turns and the wait period. Finally, despite a tendency to lower activity during both movement periods (*Figure 2G–H*), the data does not show a statistically significant difference on average activity during the wait period compared to the Return and Go periods.

## Population analysis of the EPN reveals its coding of reward, left and right but not difficulty

As a first objective, we sought to analyze coding of stimulus difficulty and reward, given previous reports that GPi/EPN neurons code for the value of expected positive and negative outcomes (*Stephenson-Jones et al., 2016*) as well as the spatial position of the expected positive and negative outcomes (*Hong and Hikosaka, 2008*). We began by trying to understand how these variables were represented in the population activity. For population level analyses throughout the study, we pooled recorded units from all animals to construct a pseudo-simultaneous population. We calculated the instantaneous variance associated with these two variables, as well as to left and right trials (*Figure 3A*).

In agreement with *Figure 2G*, there is a high variance associated with correct and incorrect trials during the evaluation period of the task. Further, left/right variance is increased during the turning period. In concordance with *Figure 2G*, the variance associated with difficulty (stimuli) is remarkably low.

Next, to understand the population dynamics associated to these task parameters we performed demixed Principal Component Analysis (*Kobak et al., 2016a*). This allowed us to generate a low-dimensional space that captures variance related to specific task parameters: condition-independent temporal dynamics, correct and incorrect trials, stimulus difficulty, and side (left and right) of turning. We hypothesized that condition-independent temporal dynamics would include kinematic relationships while marginalizations in correct/incorrect would include reward signals; the left/right marginalization would show angular velocity-related activity, and the difficulty marginalization (that is, the segregation related to the deltaFreq of the auditory stimuli) would show a gradation related to the action value.

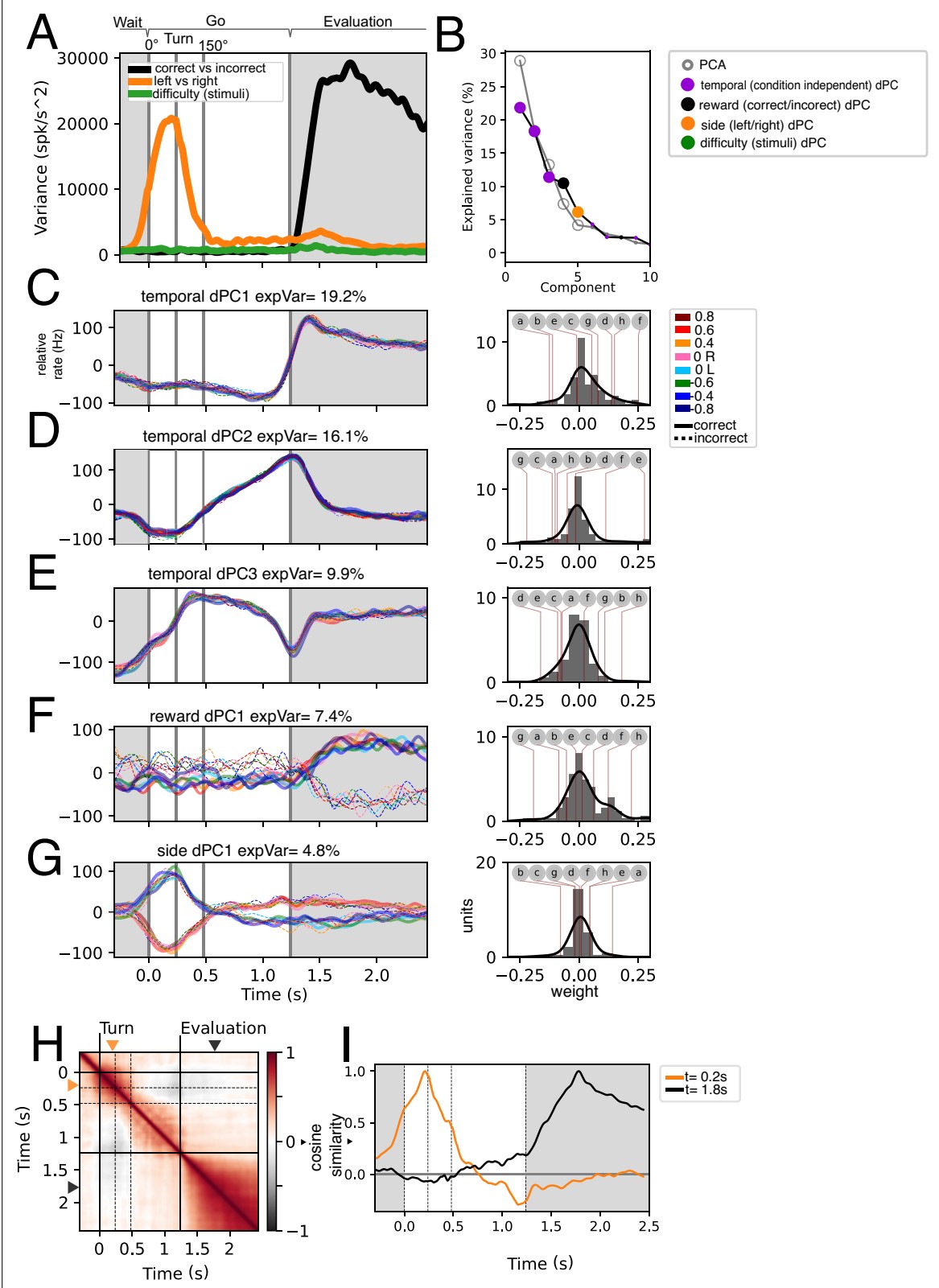

**Figure 3.** Stimulus, difficulty, reward, and temporal population dynamics during Go trajectories. (**A**) Instantaneous variance associated with trials segregated by left-right, correct/incorrect, and stimuli identity. (**B**) Principal Components and demixed Principal Components sorted by explained variance. Note that the difficulty demixed Principal Component Analysis (dPCA) does not appear within the 10 first components. Statistically significant PCs (n=5) were signaled by a bigger dot. (**C–G**) First five dPCs sorted by variance are shown: Temporal (condition-independent), reward, and side. Note

*Figure 3 continued on next page*

*Figure 3 continued*

that for traces in C-E t s (n = 16). Weights for each individual dPCs are shown to the right. Lettered circles on top are individual units' weights shown on **G**. (**H**) Population self-similarity across time. Cosine similarity was calculated for population vectors on the de-meaned (temporal dynamics removed) population activity (see Methods). (**I**) Population vector similarity across time for two time points: t=0.2 s and t=1.8, which correspond to the moments of peak side and reward variance, respectively.

The online version of this article includes the following figure supplement(s) for figure 3:

**Figure supplement 1.** Example units.

**Figure supplement 2.** Stimulus presence, irrespective of identity, and angular velocity best explain entopeduncular nucleus (EPN) activity around turning epoch.

After performing dPCA and sorting the extracted components by the explained total variance (ETV), we find that the first three are condition-independent components (temporal). Further, only five PCs are statistically significant (see Methods). The most prominent dynamic, temporal dPC1 which explains 19.2% of total variance, separates the outcome period from the wait-and-go periods akin to a step function. The right panel in *Figure 3C* shows the weights of individual units onto this dynamic. We show individual example units in *Figure 3—figure supplement 1* which correspond to the letter markings in the weight panels (*Figure 3C–G*, right panels). The second temporal dPC (16.1% ETV) exhibits ramping activity during the Go moving period (*Figure 3D*), which could be encoding time or the distance to goal. The third temporal dPC projection exhibits a triphasic dynamic during the moving period, potentially correlating with the speed of the animal's trajectory (*Figure 3E*).

The fourth and fifth dPCs correspond to correct/incorrect and side, respectively. As expected, the correct/incorrect dPC separates correct from incorrect trials during the evaluation period (*Figure 3F*). Further, the side dPC is maximally different during the turning period (*Figure 3G*). Note that the weights of these axes are positive and negative centered around zero. This means that units encoding reward can code a rewarded trial as an increase in activity or as a decrease in activity and vice versa (example units are shown in *Figure 3—figure supplement 1g and h*). Similarly, leftwards turns can be encoded as a decrease in activity or an increase in activity (*Figure 3—figure supplement 1a and b*).

Interestingly, the marginalization of difficulty (stimuli) did not yield a significant dPC. This, coupled with the comparatively low difficulty-associated variance (*Figure 3A*) suggests that difficulty coding is not well represented in this dataset. Still, by comparing correct and incorrect trials (with stimulus) vs false alarm trials (without stimulus), some units show stimulus-associated activity (*Figure 3—figure supplement 2A and B*). We looked to further assess whether stimulus-related activity could represent difficulty. We asked whether activity around stimulus and turning could be representing the presence of stimulus, the difficulty of the stimulus, what the animal interpreted from the stimulus (correctly or incorrectly turning to either right or left), or simply the angular velocity (*Figure 3—figure supplement 2C*). Given their overlap, we calculated the delta R2 of each of these possibilities, which measures how much the goodness of fit improves when considering each variable versus only the others (see Methods). This analysis shows that including information about the difficulty or the interpretation are worse at explaining activity than the sole presence of stimuli (*Figure 3—figure supplement 2D*). Further, the variable that best describes the activity in this period is angular velocity.

To assess how side and reward coding might correlate/covary with each other, we computed the population self-similarity across time (*Figure 3H*). This self-similarity was computed from the demeaned responses, which eliminates condition-independent (temporal) coding. We found that the evaluation period is well separated from the Go period and the interaction between them is very low. We assessed the similarity of the population vector at the time points of greatest side and reward variance (t=0.2 s and t=1.8 s, respectively; see *Figure 3A*) against the rest of the time bins (*Figure 3I*). This shows that turning and reward coding are largely uncoupled.

The presented analyses in *Figure 3*, *Figure 3—figure supplement 2* show that at the population level the EPN recorded activity in freely moving mice encodes reward and turning side, both of which are encoded positively and negatively. Minimal contribution of difficulty was detected.

## Contextual motor coding across different periods

As a second objective, we analyzed the relationships between the activity of the recorded units and kinematic variables. Since one of the most prominent population dynamics describes a linear relationship between time/space and population rate (*Figure 3D*), we sought to study spatio-temporal

variables as well. By design, the task has two distinct epochs of displacement: Return trajectories (to the waiting corner) and Go trajectories (to either of the reward corners). We hypothesized that purely 'motor coding' would be invariant to these two contexts.

*Figure 4A–C* shows three example units. Each unit has two raster plots corresponding to Return and Go trajectories, aligned to movement start and sorted by movement duration. We grouped and averaged their activity into six bins for visualization and presented the average firing rate and average speed. All units presented have significant correlations with at least three of the variables presented, which suggests that a multiple regression model is better suited for the analysis.

For the construction of the multiple regression models, we divided variables into kinematic (measured from the body of the animal) and spatio-temporal (measured in relation with the arena) and reward-related (*Figure 4*, *Figure 4—figure supplement 1*). Given the possibility of collinearity between measured variables (*Figure 4—figure supplement 1*), we employed LASSO regression which introduces a regularization term that shrinks the coefficients and penalizes colinear variables, acting as a feature selection (see Methods). To assess how well the model explained the neural activity we calculated a 10-fold cross-validated R2 for each unit, which ensures that no variance is predicted by chance.

The group of kinematic variables is worse at explaining single-unit activity when compared to spatio-temporal and reward-related variables (*Figure 4D*). However, a model incorporating all three groups of variables outperforms any specific category.

An important goal of this study was to compare the encoding kinematic and spatio-temporal variables between the Return versus the Go trajectories (two periods of similar movement but different sub-goals).

By fitting models separately for the Return and Go trajectories, we found that the average cvR2 for models fitted with Go trajectory data is higher for kinematic, and spatio-temporal variables, as well as for a mixed model (*P*<0.001, Wilcoxon test; orange and green bars in *Figure 4E*). This implies both kinematic and spatio-temporal variables are best represented during the trajectories that animals performed towards the reward ports (Go trajectories). This is probably partly due to units that are more responsive during the Go trajectory (*Figure 2E*).

Next, we sought to investigate which variables were most relevant for the models' success. To this end, we constructed single-variable models for each unit and plot the mean cvR2 for each of the measured variables (*Figure 4F*). This was also done separately for return (orange) and go (green) trajectories, as well as for the entire duration model (gray). Amongst the kinematic variables the best variable for explaining the neural data variance was the body speed, followed by the head speed and the angular velocity (*Figure 4F*). We found that relationships with tested variables are not generalizable between both contexts. Model performance is reduced drastically (*P*<0.001, Wilcoxon test) when switching the test data to that of the other context (*Figure 4G*). This is unlikely to be due to over-fitting since models were trained through cross-validation. The implication of this finding is that understanding the relationships between kinematic and spatio-temporal variables in one context is not enough to predict how the population will respond in another context.

To further examine why these models were not generalizable between contexts, we calculated the partial correlation coefficient for both contexts for three of the variables: distance to goal, speed, and angular velocity (*Figure 4H–J*). We fitted the full model minus the variable in question to the data of each of the two contexts and calculated the correlation coefficient with the residuals (see Methods). We plotted the return trajectory correlation coefficient (x-axis) against the go trajectory (y-axis). In a hypothetical scenario where EPN units reflected purely motor coding, all data points would lie within the identity line (dashed line). However, this analysis reveals that some units display a significant partial correlation with the variable during either Return trajectories (orange dots) or the Go trajectories (green dots). Even though there are units that display a significant correlation in both trajectories (brown dots), these units do not solely lie within the I and III quadrants of the cartesian plot, which means that some units have statistically significant correlations with opposite signs during distinct trajectories (*Figure 4H–J*, pie charts). Indeed, for angular velocity and body speed only about a third (31%) have significant correlations in both trajectories, but only about a fifth of all units have significant correlations with congruent signs (*Figure 4H–J*, pie charts). Overall, these analyses paint a complex picture, where the subset of units that encode each of the variables studied in one context is not the same as those that encode the same variable in the other context. Further, the differing magnitude

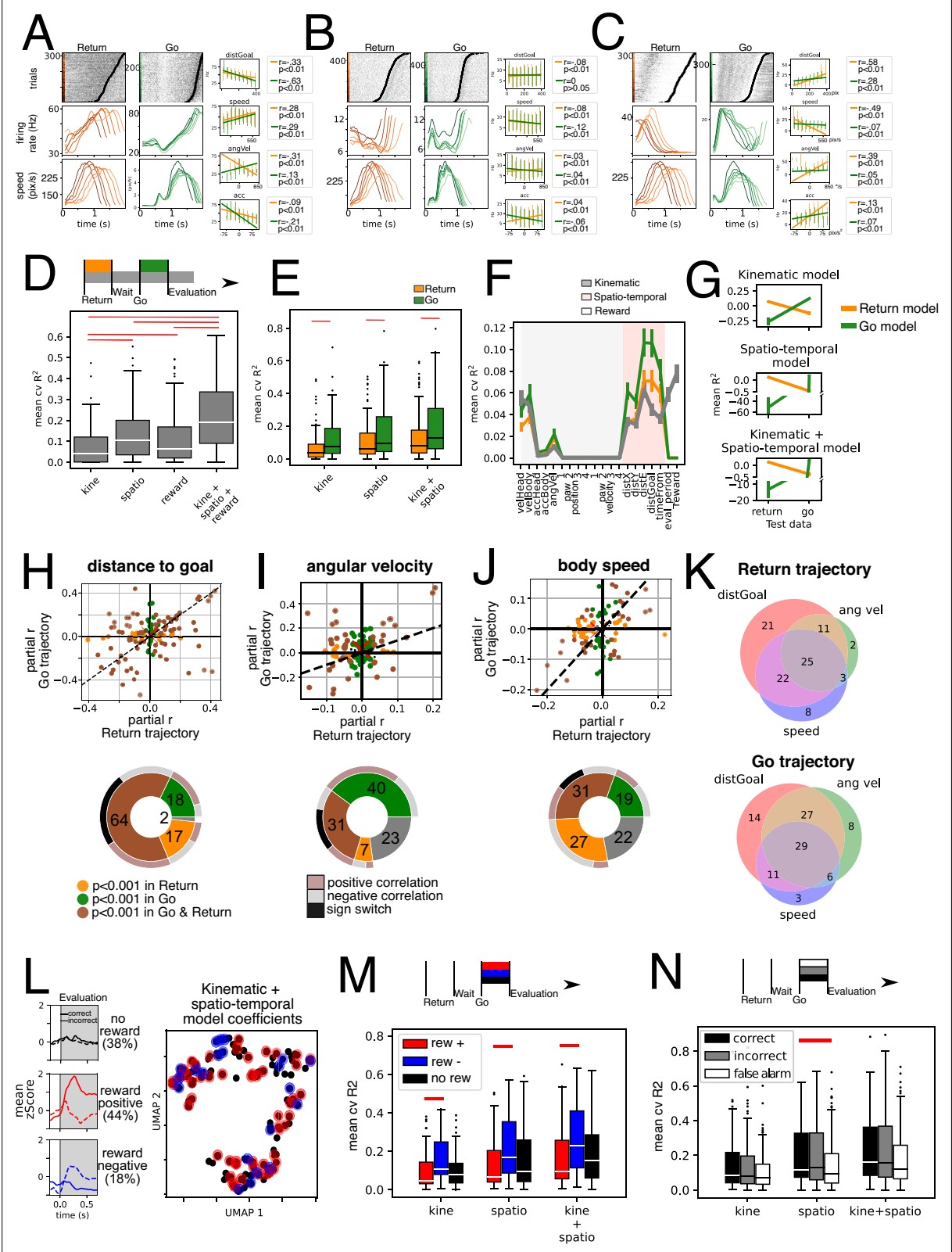

**Figure 4.** Kinematic and spatio-temporal coding during Return and Go trajectories. (**A–C**) shows three example units. Raster plots show activity for return (left) and Go (right) trajectories sorted by trajectory duration (black circle). They were then grouped into six traces, and average firing rate is presented below. Also, the average speed of the animals locomotion is presented below. Simple linear regression for four different variables presented on the right panels, for Return (orange) and Go (green) trajectories separately. (**D**) Average 10-fold cross-validated R2 for multiple regression models

*Figure 4 continued on next page*

*Figure 4 continued*

fit on data from the entire trial duration (return, wait, go, and evaluation periods) for kinematic, spatio-temporal and reward-related variables, as well as full model fit on all these variables (n = 118 units). Red line represents a statistically significant difference (p<0.01, Wilcoxon signed-rank test). (**E**) Average 10-fold cross-validated R2 for models for multiple regression models fit separately on Return (orange) and Go (green) trajectory data. Red line represents a statistically significant difference (p<0.01, Wilcoxon signed-rank test). (**F**) Mean cvR2 for single variable models. Shading corresponds to kinematic (gray) and spatio-temporal (pink) variables. Reward-related variables are unshaded. Models were fit with Return (orange), Go (green) and entire trial duration (gray) data, n = 118 units. (**G**) Trained models for Return and Go periods in E were tested using Return and Go data. Individual models for kinematic, spatio-temporal, and a mixed kinematic and spatio-temporal model were tested. Note all models tested on data not used for training were significantly lower than when using training data (p<0.01, Wilcoxon signed-rank test), n = 118 units. (**H**) Upper: Partial correlation coefficient (**r**) was calculated for the distance to goal variable by fitting residuals of model on kinematic variables (excluding spatio-temporal variables). This was done separately for Return (x-axis) and Go (y-axis) data. Dashed line corresponds to the identity line (x=y). Lower: Pie chart summarizing data above. Brown: Units with statistically significant correlations with the distance to goal variable in Go and Return periods. Green: units with statistically significant correlation only during the Go period. Orange: units with statistically significant correlation only during the Return period. Outer pie summarizes the sign of the correlations where gray and salmon represent negative and positive correlations, respectively, in both Return and Go periods. Black represents units that switch signs in Return and Go periods. (**I**) Upper: Partial correlation coefficient of residuals of a model with all spatio-temporal and kinematic variables except angular velocity, separately for Return and Go trajectories. Lower: similar to H. (**J**) Upper: Partial correlation coefficient of residuals of a model with all spatio-temporal and kinematic variables except body speed, separately for Return and Go trajectories. Lower: similar to H. (**K**) Venn diagrams of percentage of total units with a significant partial correlation coefficient for the variables distance to goal, angular velocity and body speed for Return and Go trajectories. (**L**) Left: Mean zScore activity of units segregated according to activity during reward. Top panel shows mean of units unresponsive to reward. Middle and lower panel shows average activity of units with statistically significant difference between rewarded and unrewarded trials in a 250ms window after head entry into the lick port. Middle panel corresponds to reward-positive units that showed an increase in firing rate with rewarded trials (auROC >0.5, p<0.001 permutation test) while the lower panel corresponds to reward-negative units that showed a decrease in activity with rewarded trials (auROC <0.5, p<0.001 permutation test). Right: Dimensionality reduction performed on the regression coefficients on a model fitted Return and Go data using kinematic and spatio-temporal variables. Red dots represent reward-positive units (rewarded trials with higher firing rate that unrewarded ones) and blue ones represent reward-negative units (rewarded trials with lower firing rate). (**M**) Mean cross-validated R2 of models fit on kinematic, spatio-temporal and both types of variables, segregated by reward coding of units (segregated as described in L). Red lines indicate statistically significant difference (Mann-Whitney U test, p<0.01), reward positive n=52 units, reward negative n = 21 units, reward insensitive n = 45 units. (**N**) Mean cross-validated R2 of kinematic, spatio-temporal, and mixed models. Data was segregated according to trial type: correct, incorrect, and false alarm, n=118 units per condition. Panels A-C inserts, error bars are shown in place of individual points (200ms bins), however for regressions the individual datapoints were used.

The online version of this article includes the following figure supplement(s) for figure 4:

**Figure supplement 1.** Variables used to fit models.

of the correlation explains why models trained on one trajectory are not generalizable to the other trajectory. Some extreme cases of this instability of coding are those units that have a sign switch between one context and the other (black outer pie in *Figure 4H–J*).

Given that we found that one unit might significantly encode multiple variables, we wished to investigate how the significant partial correlations for these three variables might be distributed amongst the recorded units. In *Figure 4K* we plot Venn diagrams of the percentage of units that significantly code each of these three variables during the Return (top) and the Go (bottom) trajectories and their overlap. We find that there is a similar landscape in both periods of movement wherein most units have significant correlations with more than one variable. However, as examined before, the subset of units in each of the categories is not the same from one context to the other.

## No segregation of units coding for kinematic, spatio-temporal variables and reward

Next, we asked whether the recorded units could be grouped based upon what variables they code. That is, we sought to understand whether groups of units might encode some variables more strongly than others. Previous reports suggest that EPN units that project to the lateral habenula encode reward as a decrease in firing rate. Thus, we wished to ask whether reward encoding units can code kinematic and spatio-temporal variables as well.

To this end, we first segregated units upon their reward coding properties: reward-positive (which increased activity with reward) and reward-negative units (which decreased activity with reward). We performed auROC on the 250ms after head entry comparing rewarded trials and incorrect trails (p<0.001, permutation test). Mean activity of reward-insensitive, positive, and negative units is shown in *Figure 4L*. Next, we performed a dimensionality reduction on the coefficients of the model that best explained both contexts (kinematic + spatio-temporal model on pooled data) using UMAP

(*McInnes et al., 2018*). We observe a continuum rather than discrete clusters (*Figure 4L*). Note that individual units are color coded according to their responsivity to reward. We did not find a clear clustering either. This implies that, overall, coding for kinematic and spatio-temporal variables is uniformly distributed and the coding of these variables does not segregate the coding for reward. Interestingly, reward-sensitive units encoded kinematic and spatio-temporal parameters similarly to reward-insensitive units (*Figure 4M*). Reward-positive units are better at encoding these variables than are reward-negative units. However, neither type of reward-sensitive units had a significantly different cvR2 than that of reward-insensitive units.

To ask whether action value might be influencing the way units encoded variables, we segregated trials by their outcome into correct and incorrect. Additionally, we included false alarm trials, which animals performed in the absence of any auditory stimulus. We fitted the models independently for each subset of data. We found that false alarm trials have a statistically significantly lower cvR2 only for the spatio-temporal model, not so for the kinematic one. Thus, time/space

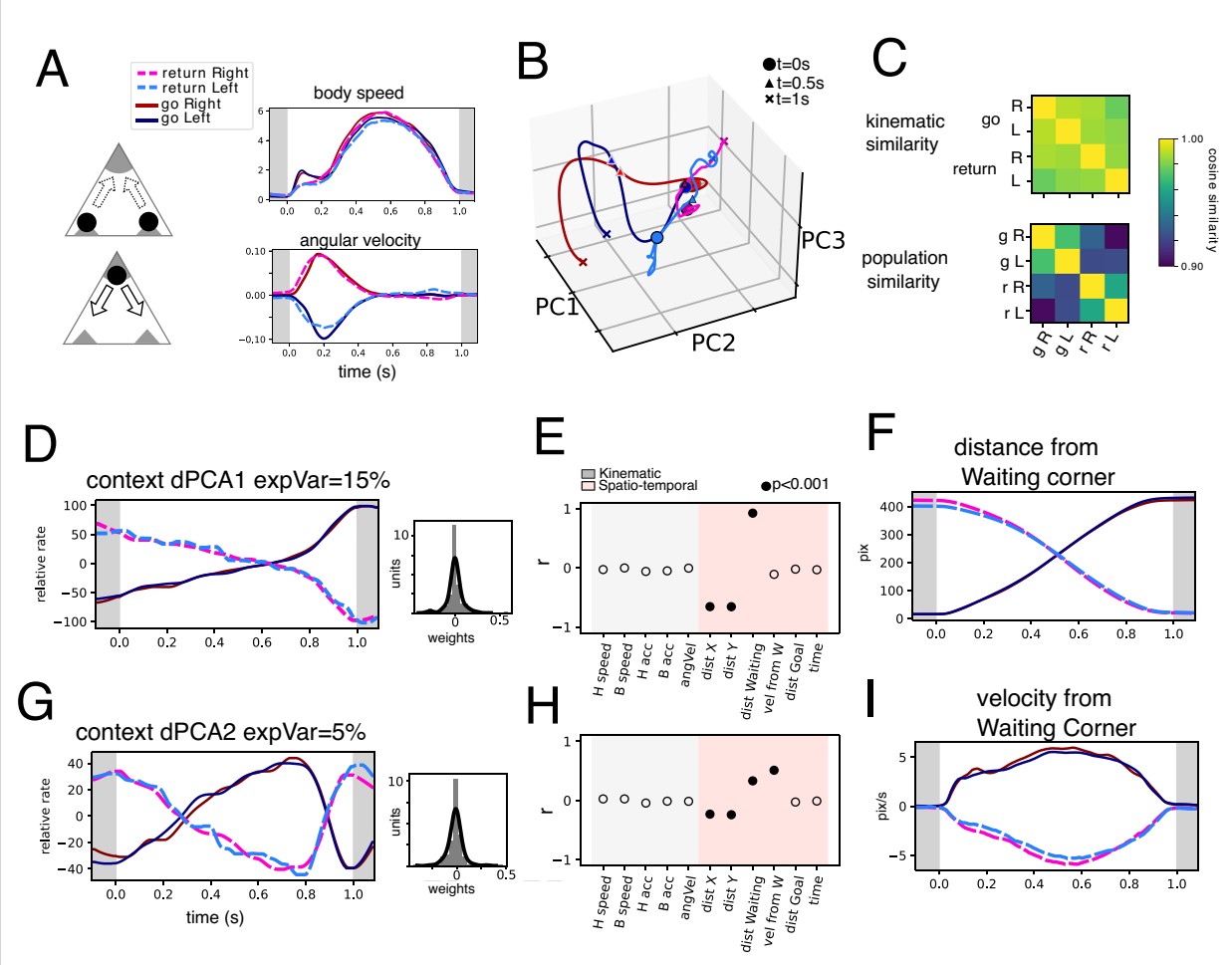

**Figure 5.** Return and Go trajectories population dynamics. (**A**) Highly similar trajectories were found for Return and Go periods based on body speed and angular velocity (see Methods). Average body speed and angular velocity are plotted for the four conditions found. (**B**) Population trajectories corresponding to the four conditions shows in A projected onto three PCs. (**C**) Cosine similarity of the four conditions based on kinematic parameters (body speed and angular velocity) and population activity. (**D**) Context dPC with most explained variance. Histogram of weights is presented on the right. (**E**) Correlation coefficient between the extracted population dynamics in D and several variables calculated for the same periods. (**F**) Distance from Waiting corner calculated for the four conditions. (**G**) Context dPC with second most explained variance. Histogram of weights is presented on the right. (**H**) Correlation coefficient between the extracted population dynamics in G and several variables calculated for the same periods. (**I**) Velocity from Waiting corner calculated for the four conditions.

The online version of this article includes the following figure supplement(s) for figure 5:

**Figure supplement 1.** demixed Principal components for Return and Go trajectories, continued.

varying signals are attenuated during False Alarm trials (the example unit in *Figure 2E* exhibits this behavior).

## What makes return and go trajectories different at a population level?

To address this question, we next sought to understand what population level dynamics are shared between Return and Go trajectories, and what underlying dynamics might be explaining context differences. To this end, we found highly similar Return and Go trajectories from a motor perspective (see Methods) based on body speed and angular velocity (*Figure 5A*). Intriguingly, by projecting the population activity (n=118 units) onto the first 3 PCs (*Figure 5B*), Return and Go trajectories occupy different subspaces in this graph (*Figure 5—figure supplement 1A and B*). Indeed, while the kinematic similarity is high between Return and Go trajectories (*Figure 5C*, upper), population similarity exhibits two groups based on the context (*Figure 5C*, lower).

To find a more interpretable dynamic that may account for the differences between the two contexts we employed dPCA, marginalizing by context, left-right, and temporal components. We hypothesized that temporal components shared by both contexts might exhibit kinematic variables and that left-right marginalization would encompass angular velocity. After performing the dPCA procedure (see Methods), we sorted the components by variance (*Figure 5—figure supplement 1C*). Note that after marginalization upon the desired variables, the extraction of the dynamics is unsupervised. This is to say that extracted dynamics do not have to necessarily produce traces related to individual kinematic parameters, and they are unlikely to do so given the multiplexing of encoded variables (*Figure 4K*). However, we sought to interpret the resulting traces based on measurable variables. Thus, we compared the resulting trace with the variables measured on the same time periods by calculating the correlation coefficient and testing significance through permutation testing (p<0.001).

The contextual marginalization produces two significant dPCs that explain the differences between Return and Go trajectories. Indeed, the contextual dPC1 explains 15% of the total variance (*Figure 5D*), and Return vs Go traces exhibit oppositely evolving temporal dynamics. The spatial variable of distance from the waiting corner best explains these contextual traces, while no kinematic variable correlates with these traces. Interestingly, context dPC2 (*Figure 5D*) exhibits a high similarity to the speed-correlated temporal dPC2 (*Figure 5—figure supplement 1E*). However, context dPC2 shows oppositely evolving dynamics symmetric over time between Return and Go trajectories. This suggests that there is a simultaneous representation of speed and a spatially referenced velocity (*Figure 5G–I*). Indeed, context dPC2 correlates well with the velocity from the waiting corner.

Condition-independent (temporal) dPCs summarize population dynamics shared by both Return and Go contexts (*Figure 5—figure supplement 1*). Temporal dPC1 has a high correlation with distance to goal (*Figure 5—figure supplement 1D*) while temporal dPC2 (*Figure 5—figure supplement 1E*) is well correlated to the speed. Note that these traces do not correlate with one specific variable, which is to be expected given the previously established high level of multiplexing (*Figure 4H–K*). The marginalization for side (side dPC1, *Figure 5—figure supplement 1G*) produces traces that have a significant correlation with angular velocity, both for the Return and Go trajectories. Note that Go trajectories occurred under the presence of an auditory stimulus, and not the Return trajectories; however, traces corresponding to left and right turns (irrespective of goal) show good overlap. Thus, this further confirms that the associated differences between left and right turning are well explained by angular velocity. We conclude that context-dependent dynamics explain variables whose measurement depends on specific places in the arena. Thus, spatially biased variables have an important representation of the EPN population activity.

## EPN units are modulated by gait and licking

Our findings reveal a robust representation of whole-body speed (*Figure 4*) as well as spatially biased variables (*Figures 4–5*). However, prior studies have indicated a somatotopic representation in the GPi across various species [humans, non-human primates, cats *Baker et al., 2010*; *Larsen and McBride, 1979*; *Nambu, 2011*]. Therefore, we examined the EPN's modulation by individual movements of the paw and tongue, using the cyclic nature of gait and licking to examine the potential presence of muscle-level activity since they exhibit a clear flexor-extensor muscle relationship.

Given that animals were recorded from a bottom-up view, we were able to track individual paws of the animal (*Figure 6*). We searched for Return and Go trajectories with at least four individual strides

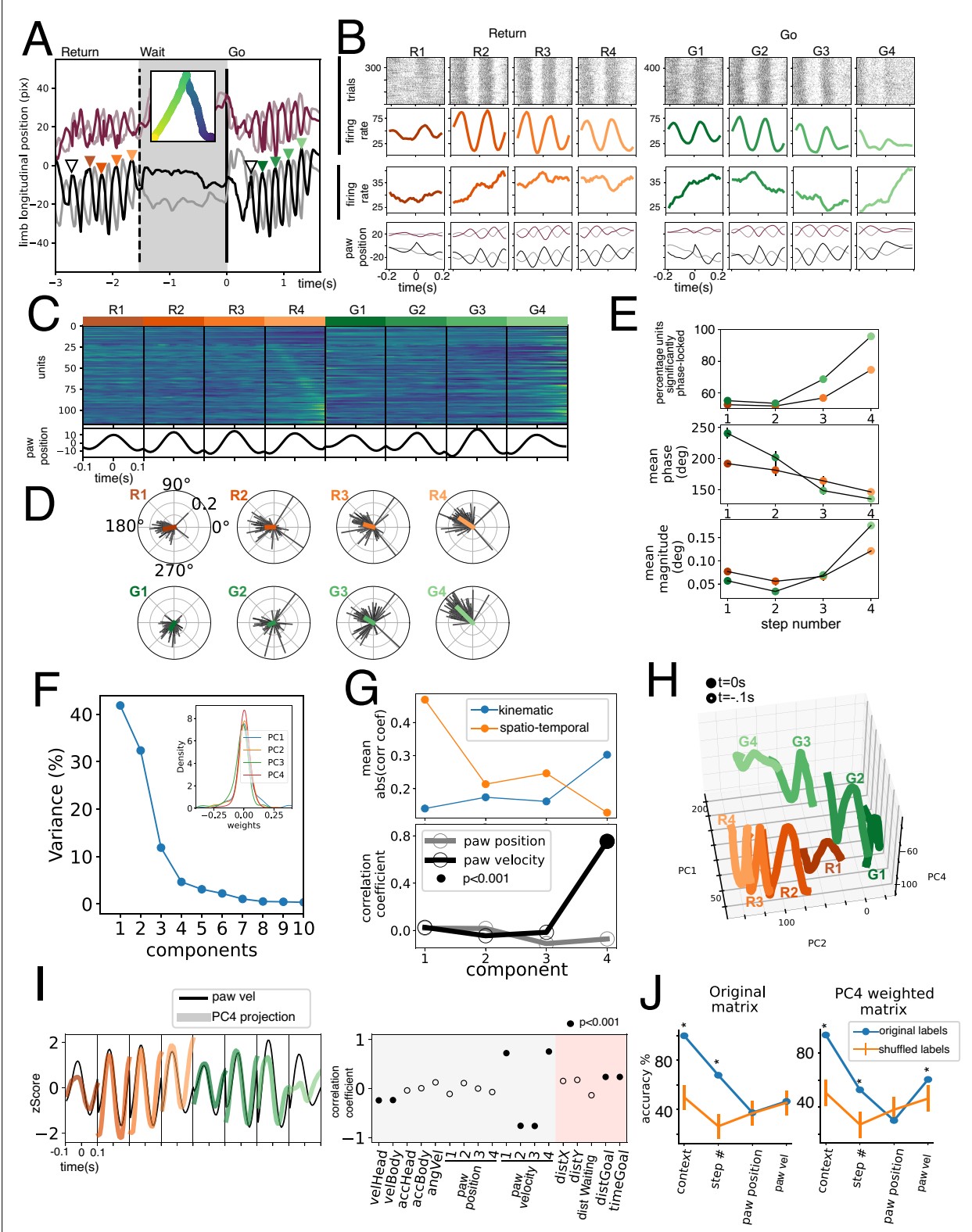

**Figure 6.** Gait cycle modulation of units. (**A**) Longitudinal position of four paws of a mouse during a return, wait, and go periods. Inset shows body position in time. Colored triangles show identified events for unit alignment. (**B**) Activity of two simultaneously recorded example units in the eight strides studied. Lower row shows average paw position corresponding to the trials shown above. (**C**) Population activity during the four strides. (**D**) Spike gait-phase average vector for all significantly modulated units during eight strides. (**E**) Upper: Fraction of modulated units (with a significant directionality Rayleigh test p<0.001). Middle: Mean phase of population vector. Lower: Mean population vector magnitude. (**F**) Variance explained

*Figure 6 continued on next page*

*Figure 6 continued*

by Principal Component Analysis (PCA) decomposition of data shown in C. Inset shows first four PCs' weight distribution. (**G**) Upper: mean absolute correlation coefficient for kinematic and spatio-temporal variables for the first four PCs. (**H**) Population projection of data in C onto PC1, PC2, and PC4 for the four Return strides (in orange tones) and the Go strides (in green tones). (**I**) Left: PC4 population projection for the eight strides studied plotted on top of contralateral hindpaw velocity. Right: correlation coefficient with individual kinematic and spatio-temporal variables. Filled circles represent a statistically significant correlation (permutation testing). (**J**) Population Support Vector Classifiers were trained for different variables (x-axis). Accuracy was assessed for the classifier trained on the original labeled population vectors (in blue) and for vectors trained on shuffled labels (in orange). *p<0.001, permutation testing. Note that paw velocity could not be decoded from the original matrix but could be decoded from the PC4 weighted matrix.

(*Figure 6A*, triangles) and used them to align unit activity. *Figure 6B* shows two example units simultaneously recorded from different channels. Note how Unit 1 has a very robust stride-locked activity, while Unit 2 shows a smaller stride-based modulation and instead becomes immersed in an evolving temporal dynamic. The average paw position for the session (*Figure 6B*, bottom) is well preserved, and that there is a high level of correlation or anticorrelation, which justifies our using only one paw to study. This procedure was performed for all units studied and is presented in *Figure 6C*, sorted by peak activity during the step 4 in Return trajectories. This sorting is only slightly preserved in the rest of the steps, highlighting the dynamic nature of this gait modulation.

Since locomotion necessitates a coordinated contraction of flexor and extensors during each stride, we hypothesized that if individual muscle activity was represented in the GPi, we would find stride-cycle locked activity in symmetric opposing phases (flexors vs extensors). We calculated the spike- contralateral hindlimb phase locking for each unit at each step studied. In *Figure 6D* we plot the average vectors of individual units with significant phase locking (p<0.001, Rayleigh directionality test) for each of the eight steps studied. Note how for each step units tend to point to one phase and not to the antiphase. Further, there is a shift in the mean phase of the paw (*Figure 6D–E* middle), for Return and Go trajectories. Surprisingly, as the step number increases (which corresponds to steps closer to the goal of the trajectory), a greater fraction of units has a significantly stride phase-locked activity (*Figure 6E*, upper panel). The magnitude of the average vector also increases with step number (*Figure 6E*, lower panel), suggesting an increase in phase-locking. These results are not supportive of this activity being muscle related, since most gait-modulated units are biased toward a single phase of gait, with no antiphase representation (*Figure 6C*). Further, gait modulation increases with steps closer to the goal, and phase precession would not occur if units were strictly related to muscle activity.

Given the widespread existence of this phase-locked activity, we hypothesized the presence of a covariance plane that would encompass it. Thus, by performing PCA on the population activity (*Figure 6C and F*) we could understand what variable might be represented at a population level. We found that the population projection onto the first three PCs correlated best with spatio-temporal variables (*Figure 6G*, upper panel) while the fourth PC had a good correlation with paw velocity (*Figure 6g*, lower). Indeed, there is a clear overlap of PC4 projection with paw velocity (*Figure 6I*). Finally, we sought to compare the robustness of the representation of the different variables on the population activity. To this end, we trained linear support vector classifiers to discern between Return and Go trajectories (context), step number, and paw position and acceleration (*Figure 6J*). Paw velocity could only be decoded above chance on the PC4 weighted matrix, not on the original data. These results support the existence of a relatively weak representation of the paw velocity at a population level. Paw-related PC4 captures only 5% (*Figure 6F*) of the variance of this matrix, which includes less time compared to other population analyses (*Figures 3 and 5*), does not segregate by turning direction, and does not include a reward period.

A final test of kinematic representation was performed by analyzing the relationship between EPN activity and licking behavior. Individual licks could be recorded by a custom-made lickometer (*Figure 7A*, see methods and *Video 2*). We found a correlation between licking behavior and EPN activity at two different timescales: as a whole lick bout instance and at the single lick behavior. *Figure 7B and C* are raster plots of licks and a unit activity recorded simultaneously, for left and right licking ports. Because of the apparent similarity between lick and firing rate traces, we sought to assess whether units might be encoding licking rate during the licking bout. To this end, we computed the rescaled activity to the average duration of licking bouts. *Figure 7D*, upper panel shows the rescaled activity of three simultaneously recorded units and the corresponding licking rate. Despite

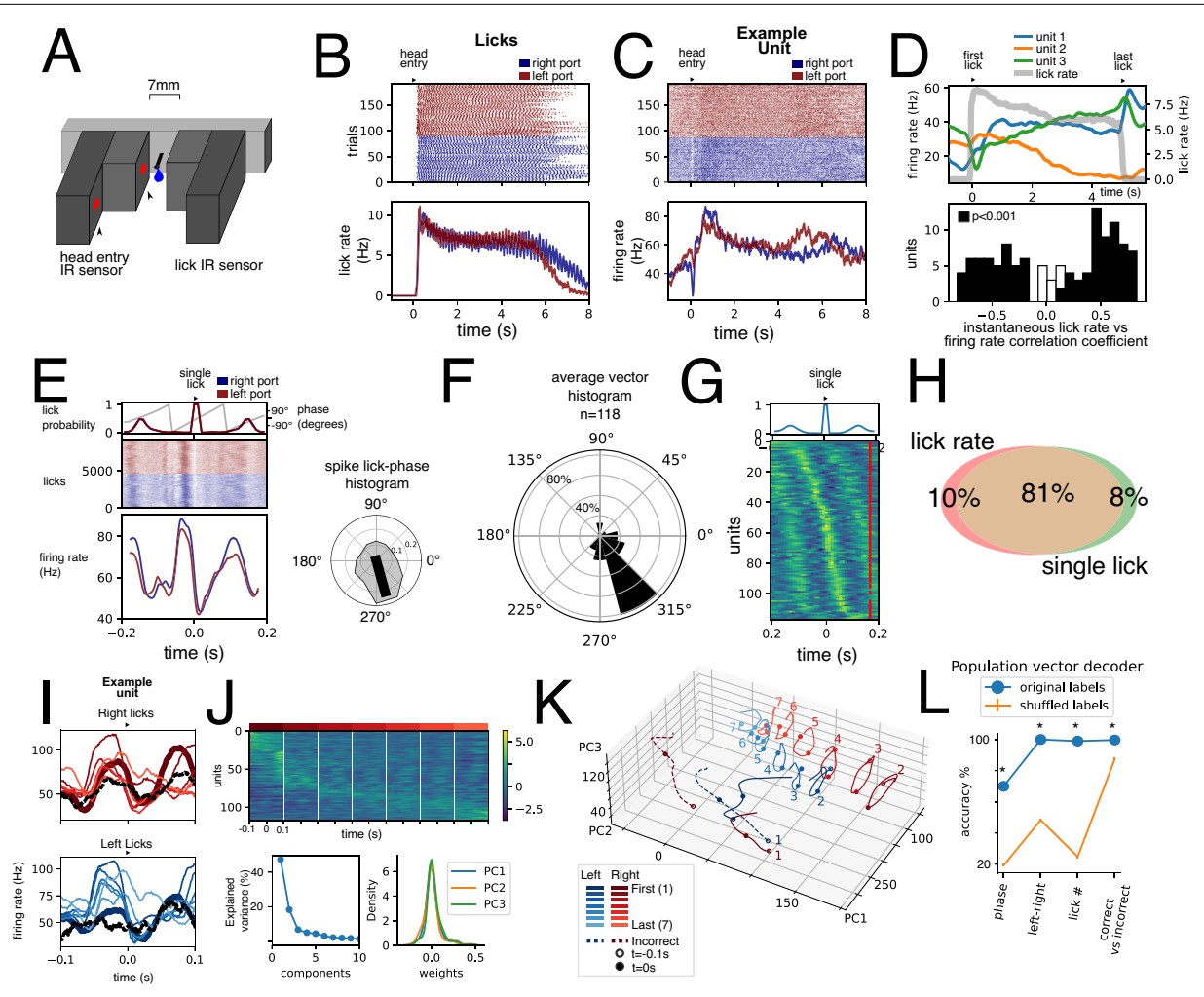

**Figure 7.** Licking behavior modulation of units. (**A**) Infrared sensor-based lickometer. (**B**) Licks of aligned to port entry for left (red) and right (blue) correct trials. (**C**) Example unit activity aligned to the same port entries as in B. (**D**) Upper: Three simultaneously recorded example units' (blue, orange, and green) activity rescaled to average lick bout duration. Lick rate is shown in gray. Lower: Histogram of Pearson correlation coefficient between instantaneous firing rate and instantaneous lick rate for all units. Black bars represent significantly correlated units (p<0.001, permutation test). (**E**) Example unit shown in C aligned to single left (red) and right (blue) licks. Upper inset shows the probability occurrence as well as the lick cycle phase derived from the lick probability. Right inset shows the spike lick-phase polar histogram for same unit. Black line represents resulting vector with statistically significant directionality (Rayleigh test, p<0.001). (**F**) Polar histogram (bin = 30°) of average vector phase direction. Polar axis is percentage of units. (**G**) Average single lick activity for all recorded units sorted by peak activity. Upper inset: average lick probability. (**H**) Venn diagram showing percentage of units of the total population (n=118) of units with statistically significant correlation with lick rate (Pearson correlation coefficient, p<0.001, permutation test) and/or with single licks (Rayleigh directionality test, p<0.001). (**I**) Single lick average activity segregated by first and last lick, five interspersed licks, and incorrect lick for left and right licks for the same unit shown in C and E. (**J**) Population activity for n=118 units for seven right licks sorted by lick with most activity. Lower: fraction of variance explained by Principal Component Analysis (PCA) decomposition PCA weight for first three components. (**K**) Projection of lick population activity for the 16 licks considered onto first three PCs. (**L**) Population vector decoders for different variables with original labels (blue) compared to population vector decoders with shuffled labels (n=1000). *p<0.001, permutation test.

varying dynamics exhibited by these three units, all three have a high degree of correlation with the lick rate. *Figure 7D*, lower panel shows the correlation coefficient between the instantaneous firing rate and the instantaneous lick rate within a lick bout for all units. Indeed, most units (91%) exhibit a positive or negative significant correlation with lick rate (p<0.001, permutation test). We then obtained PETHs of activity related to single licks in the left and right lick ports (red and blue, respectively), as shown in *Figure 7E* for the same unit as in *Figure 7C*. By extracting the lick phase from the lick probability (sensor crossings, *Figure 7E*, upper), we could generate a spike – lick-phase histogram and calculate the resulting vector (*Figure 7E*, right). When performing this for all units, it was clear that

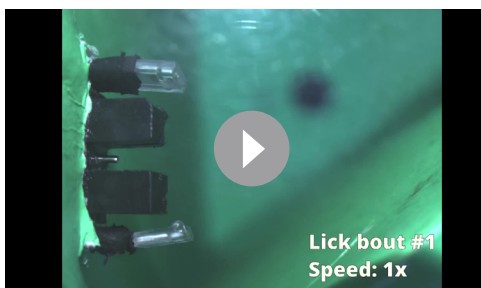

**Video 2.** Lickometer detected individual licks. Individual licks could be recorded by a custom-made lickometer. The lick port was equipped with a double infrared sensor to capture head entry and single licks separately. The lickometer IR sensor was inside a 7 mm wide slit, inside which the reward delivery spout was placed. The lickometer slit was just wide enough to fit the tongue and deep enough to evoke a clear tongue protrusion. Rewards consisted of 3 µl droplets delivered through a solenoid, which was calibrated daily. https://elifesciences.org/articles/98159/figures#video2

they fire more strongly in a specific phase of the licking behavior. This is also shown in the heatmap of perievent activity sorted by peak activity (*Figure 7G*). Thus, we show that nearly all units (99%) exhibit correlation with lick rate, single licks, or both (*Figure 7H*).

What is the significance of such a robust representation of licking by nearly all recorded units? We hypothesized that different aspects of a single lick might be represented. To this end we obtained PETH for 16 different licks per unit: the first and last licks per bout, as well as five licks spread out within the bout, right and left. We noticed that occasionally animals would execute one lick when animals performed incorrect or false alarm trials, which we termed an incorrect lick (right and left). This allowed us to analyze a similar motor output (a single lick) performed under different contextual situations. An example unit (*Figure 7I*) shows that despite similarities between all these conditions, firing rate traces exhibit clear differences. *Figure 7J* shows a heat map for seven right licks during a bout sorted by the lick with highest activity. This highlights that while there is a modulation at the single lick timescale, a whole-bout modulation is also present. After PCA dimensionality reduction (*Figure 7J*, lower), we projected the population responses on the three main principal components (*Figure 7K*). The population responses for the 16 conditions maintain a circular (cyclic) response. However, they occupy distinct spaces in this state space which suggests that the EPN population can distinguish amongst them. To assess this, we trained support vector classifiers on population vectors for lick phase, left vs right licks, lick number, and correct vs incorrect licks. All these variables could be accurately decoded above chance. Together, these results show that most units in the EPN are modulated by licking behavior at different timescales. Further, information about licking kinematics is simultaneously represented with other contextual variables.

## Discussion

This study highlights the dynamic nature of kinematic representation within the EPN. It is particularly striking that spatio-temporal dynamics are better at explaining EPN activity than are kinematic ones. Further, relationships between these variables and EPN activity fluctuate across two largely similar periods of movement (Return vs Go). What is more, units multiplex different functions like reward, whole-body movements, and individual movements (paw and licks) with contextual variables. Interestingly, we were unable to find a strong correlate of difficulty.

The principal goal of this study was to investigate how the activity of entopeduncular nucleus is related to kinematic variables, reward, and difficulty/value, particularly during locomotion. Two previous findings had a particular influence over the study design: (1) the existence of reward and value coding with a spatial bias, and (2) that kinematic coding was unstable and varies depending on cognitive states. To this end, we designed a task that involved decision confidence, reward, and two periods of displacement.

### Prominent representation of evaluation period, reward, but no difficulty in the EPN

We found that there is a robust representation of evaluation period and reward. When analyzing the population dynamics (from pooled units from all animals to construct a pseudo-simultaneous population, which assumes homogeneity across subjects) underlying the Go period, we found that a dynamic that separates the evaluation period from the rest of the task captures the most variance (*Figure 3C*).

Further, a statistically significant reward dPC was found (*Figure 3F*). Importantly, reward coding is also represented within this nucleus by reward-positive and negative neurons. These findings support previous studies (*Hong and Hikosaka, 2008*; *Stephenson-Jones et al., 2016*).

Stephenson-Jones et al. found that a subset of EPN neurons projecting to the habenula encode value of expected outcomes in a classical conditioning task. We hypothesized that by varying the difficulty of stimuli (and thus increasing uncertainty during difficult choices) we could be able to modulate the perceived value of the chosen action. However, the explained variance by this hypothesis is very low (*Figure 3*). The stimuli were able to impact the animals' response time and angular velocity (*Figure 1E–H*) which suggests some level of metacognition of the uncertainty in their decision. However, speed was not well correlated with difficulty (*Figure 1I–J*), which could imply that this metacognition diluted over time. Further, value coding was documented in neurons recorded during a head-fixed classical conditioning task (*Stephenson-Jones et al., 2016*), where the stimulus-outcome contingency might be more explicit or the coding of neurons less mixed (due to forced immobility). It is still possible that by modifying reward contingencies such as droplet size value coding could be evidenced.

In this study, we found that rewarding outcomes can be represented by EPN units through either an increase or a decrease in firing rate (*Figure 3*, *Figure 3—figure supplement 1*, and *Figure 4L*). While *Stephenson-Jones et al., 2016* found that mice lateral habenula (LHb)-projecting neurons within the EPN of mice primarily encoded rewarding outcomes by a decrease in firing rate, *Hong and Hikosaka, 2008* observed that in primates, LHb-projecting units could encode reward through either a decrease or an increase in firing rate. Thus, our results align more closely with the latter study, which also employed an operant conditioning task.

## EPN activity can represent movement but with novel considerations

In this study, we found that Go trajectories have a more robust representation of kinematic and spatio-temporal variables when compared to Return trajectories (*Figure 4E*). Indeed, several studies have found an inconsistent relationship between the GPi activity and limb movements of the primate, where units were modulated by movements under certain cognitive conditions but not others. Some hypotheses have been that GPi coding depends on movement type (ramping vs. stepped; self-paced vs step tracking, *DeLong, 1971*; *Georgopoulos et al., 1983*; *Mink and Thach, 1991*) or between identical movements performed under different cognitive states (cued vs memory-dependent *Turner and Anderson, 2005*). Further, some studies agree that self-paced movements are more weekly represented (; *Turner and Anderson, 2005*). Interestingly, while most previous studies have focused on limb movements, here we find that whole-body displacement kinematics are better at explaining EPN activity than individual limb movements. Both Return and Go movements in this study are self-paced, and they are mostly different in their relation to reward.

Representation of movements seems to be very dependent on what the movement is performed for. Kinematic and spatio-temporal variables are not stably represented across similar movements with different purposes (*Figures 4–5*). Given the sequential nature of the task, Return and Go trajectories are both required to eventually get a reward. While both trajectories have a goal: Return → auditory stimulus, Go → water droplet, they differ in that Go trajectories are performed to receive something rewarding. Indeed, we hypothesize that movements that can lead to a reward are better represented by EPN neurons. A similar conclusion was reached (*Gdowski et al., 2001*) in primates performing wrist movements, where they found that wrist movements that led to reward modulated a higher fraction of units than wrist movements that did not lead to a reward. The findings in this study differ in that kinematic relationships were indeed found in the Return trajectories, but they were attenuated.

An important finding of this study is that spatio-temporal variables are better at explaining EPN activity than purely kinematic ones, for both Go and Return trajectories. Further, these spatio-temporal variables are sensitive to action value (*Figure 4N*, false alarm trials have a lower R2 in the spatio-temporal model). Importantly, we did not find that kinematic representation was affected by action value (*Figure 4N*). Only false alarm trials had a weaker relation with spatio-temporal variables than correct trials. It is possible that value influences the spatio-temporal dynamics akin to reward prediction error, (*Dayan and Balleine, 2002*). A previous finding focusing on the decision properties of the GPi neurons found that the main correlate that the GPi has, as compared to other brain regions, was a temporal component which they called urgency (*Thura and Cisek, 2017*). This is similar to our finding

that spatio-temporal dynamics dominate EPN activity. They found that decision was not as well represented in the GPi compared to other cortical regions. This suggests that more than movement per se, the EPN might compute contextual relationships present in the task.

There is a longstanding belief that reward-related functions in the EPN are separate from motor-related functions, since anatomical studies have found two distinct projections to habenula and motor thalamus (*Hong and Hikosaka, 2008*; *Parent et al., 1999*; *Stephenson-Jones et al., 2016*; *Wallace et al., 2017*). In this study, we did not identify recorded units by their projections, however, we could evaluate their sensitivity to reward given the task design. Surprisingly, we did not find differences in the ability of reward-sensitive units to encode kinematic and spatio-temporal variables compared to reward-insensitive (*Figure 7L–M*). Given that we did not record from the entire EPN, it is still possible that another region of the nucleus might exhibit more segregation.

We found that latent population dynamics that best explain differences between the Return and Go trajectories can be best explained by spatially referenced variables (such as distance and velocity from a specific place in the arena, *Figure 5*). The similar but contextually opposite dynamics probably reflect the units switching sign between contexts that were found in *Figure 4H–J* (outer pie, black). Interestingly, a previous study in primates *Hong and Hikosaka, 2008* found that the direction of rewarded outcome greatly modulated units in a one-direction rewarded task, and switched sign when the spatial location of the rewarded target was alternated. They attributed the activity to the stimulus onset, and did not study it in regards to saccade onset. However, the direction-sensitive activity was appropriately timed to coincide with saccade performance. It is thus possible that an important function of EPN activity is to compute kinematic and spatio-temporal relations between the subject and the reward location. In this study dynamics related to (spatial) side of turning during the Go period and reward were found to be largely uncorrelated (orthogonal to each other, *Figure 3H–I*); however, both turns during the Go period (performed to get to the reward port) have similar value. On the other hand, population dynamics that explain the differences between Return and Go periods, which can be said to have different action value, did exhibit sign switching (*Figure 5D and G*). Thus, action value (moving to or away from the rewarded location) could explain these opposing dynamics. Indeed, reward prediction error models incorporate/rely on spatio-temporal variables (*Dayan and Balleine, 2002*; *Kim et al., 2020*).

Previous studies performed in humans, primates, and cats *Baker et al., 2010*; *Larsen and McBride, 1979*; *Nambu, 2011* have suggested the existence of a somatotopic organization in the GPi. We sought to test the hypothesis that this organization could be representing muscle contraction information. To this end, we analyzed two cyclic movements, gait and licking, since they require a coordinated contraction of flexor and extensor muscles. The gait coupling of EPN units has also been found in cats (*Mullié et al., 2020*). However, we found that while units did exhibit a selective firing on a specific phase of both of these cyclic movements, there was no anti-phase balance. This could imply that only flexor muscles are represented, or, more likely, that the activity is not related to muscle contraction. A further fact that is contrary to muscle level modulation of activity is that both representations, gait an licking, largely overlap. That is, units that significantly modulate to gait also do so for licking. Thus, we can conclude that in this mouse data, somatotopic organization is not present, which is a similar conclusion reached by previous primate studies (*DeLong, 1971*; *Mitchell et al., 1987*).

Finally, it was surprising to find such a robust representation of licking behavior in these units. Practically all units correlate to whole bout licking rate and/or individual licks. This implies that the same units that presented significant correlations with movement during the displacement periods, could robustly represent licking behavior. Further, the population could represent contextual parameters associated with licking (*Figure 7K–L*).

What could be the purpose of having a representation of many types of movements and variables by the same population? The data in this study suggests that action value might be key to the level of movement representation by the EPN. Cued movements (Go, correct trials) are better represented than uncued (False Alarm) ones (*Figure 4N*), which in turn are better represented than trajectories that cannot lead to reward (Return trajectories, *Figure 4E*). Further, licking, the action of consuming a reward, very robustly modulates the EPN (*Figure 7*). Thus, it is possible that the EPN serves as online feedback of the reward properties of an action.

How do the results in the present study fit in with models of basal ganglia function? Both the 'firing rate model' (*Albin et al., 1989*) and the 'dynamic activity model' (*Nambu et al., 2023*) rely on an

inhibition of basal ganglia output activity to facilitate movement. When solely considering the average population activity we did not find that movement periods exhibited a statistically significant lower activity (*Figure 2H*). In fact, reward (evaluation period of correct trials) exhibited higher population activity than the wait period despite presumably more motor output (licking) than waiting. Further, we found very complex relationships between movement and EPN activity at different timescales, which could be affected by aberrant oscillatory activity that is described in the 'firing pattern model' (*Guillery et al., 1998*).

In this study, we have focused on describing the representation of several variables in the EPN activity. However, interpreting how this complex representation affects target nuclei is further complicated by the fact that some subpopulations within the EPN form multi-transmitter synapses of excitatory and inhibitory transmission, particularly onto the lateral habenula (*Wallace et al., 2017*). In fact, simultaneous recordings of GPi and the ventral anterior thalamus, classically considered as the motor output, reveal only weak correlations (*Schwab et al., 2020*).

A weakness of the current study is the lack of characterization of neuronal subtypes. An area of opportunity for future research could be to perform photo-identification of neuronal subtypes within the EPN which could contribute to the overall description of the information representation. Further, detailed anatomical viral vector strategies could aid to improve anatomical localization of recordings, reduce reliance on histological examination, and solve some current controversies (*Lazaridis et al., 2019*).

We conclude that EPN neurons exhibit a high degree of multiplexing of diverse variables including reward, spatio-temporal, kinematic. EPN activity can reflect single movements like gait and licking as well as whole-body movements. We found that the correlation that units have with these variables is unstable and varies depending on context. Finally, spatial location of reward can influence population dynamics.

## Materials and methods

### Animals

All procedures were approved by the Institutional Committee for the Care and Use of Laboratory Animals at the Instituto de Fisiología Celular (Protocol number FTA-121–17). Universidad Nacional Autónoma de México. and the National Norm for the Use of Animals (NOM-062-ZOO-1999). Experiments used C57BL/6 J male (n=2) and female (n=4) mice (2–3 mo of age at the start of training, 6–7 mo during recordings). Animals were housed under a 12 hr light/dark cycle (lights on at 6 am) with ad libitum access to food and water before the start of behavioral experiments. For behavioral training, water was restricted so that the weight of animals was 80% of their original weight.

### Behavioral apparatus and task design

The behavioral apparatus consisted of a triangular (side = 30 cm, height = 40 cm) arena with a transparent acrylic floor with a bottom-up camera view. The apparatus was encased in a light controlled and sound attenuated chamber. One of the corners was termed the Waiting Corner and the other two were the lick port corners. The latter had lick ports. The lick port was equipped with a double infrared sensor to capture head entry and single licks separately. The lickometer IR sensor was inside a 7 mm wide slit, inside which the reward delivery spout was placed (see *Figure 7A* and *Video 2*). The lickometer slit was just wide enough to fit the tongue and deep enough to evoke a clear tongue protrusion. Rewards consisted of 3 µl droplets delivered through a solenoid, which was calibrated daily. Auditory stimuli were delivered through a tweeter that was placed outside the arena, inside the sound attenuating chamber. The task was controlled by interfacing Bonsai, Python scripts and Arduino chips.

The task involved a two-alternative forced choice. The animal was required to approach the wait corner, wait for 0.8–1.3 s after which an auditory stimulus was played. Entrance to the waiting corner was controlled by establishing a region of interest on the video feed through Bonsai. Auditory stimuli consisted of 0.5 frequency-modulated sweeps with a start frequency of 9.2 kHz and an end frequency of 0,+/-0.2,+/-0.4,+/-0.6,+/-0.8 octaves. Upwards sweeps indicated a water reward on the right lick port and downwards on the left. The animal then had to walk to the appropriate port. Correct responses were rewarded with a water drop and incorrect ones were punished with a 10–20 s timeout indicated by a dimming of the ambient light (see *Video 1*).

Training was conducted through successive approximations introducing a new rule after the previous one was acquired. To reach steady performance, animals were trained 3–4 mo, 6 d a week.

## Recordings

Movable microwire bundles (16 microwires, 32 micrometers in diameter, held inside a cannula, Innovative Neurophysiology, Durham, NC) were stereotaxtically implanted just above the entopeduncular nucleus (–0.8 AP, 1.7 ML, 3.9 DV). Post-surgical care included antibiotic, analgesic and antiinflammatory pharmacological treatment. After 5 d of recovery, animals were retrained for 1–2 wk. Unitary activity was recorded for 2–6 d at each dorsoventral electrode position and the session with the best electrophysiological (signal-to-noise ratio (>2), stability across time) and behavioral [performance, number of trials (>220)] quality was selected. Microwire electrodes were advanced in 50 micrometer dorsoventral steps for 500 micrometers in total. After experiment completion, animals were perfused with a 4% paraformaldehyde solution. Brains were extracted, dehydrated with a 30% sucrose solution and sectioned in a cryostat into 30 micron thick slices. Slices were mounted and photographed using a light microscope. Microwire tracks of the 16-microwire bundle were analyzed (*Figure 2A–B*) and only animals with tracks traversing the EPN were selected (6 out of 10). Finally, we located the final position of microwire tips and inferred the dorsoventral recording position of each of the recording sessions. Only units recorded within the EPN were included.

Electrophysiological recordings were acquired at 30 KHz (Cereplex Direct, Black Rock Microsystems, UT) high pass filtered at 750 Hz to extract spiking activity. Units with signal-to-noise ratio of at least two were sorted online through a hoop algorithm and further sorted offline (Offline Sorter, Plexon). Relevant behavioral events and video synchronization were stored as digital events.

Animal video position was extracted using DeepLabCut (*Mathis et al., 2018*). The several kinematic variables like angular velocity, velocity, and acceleration of displacement as well as spatio-temporal the relative position of the animal or time to the different corners were computed (*Figure 1C*). We identified four moments per trial to align the spiking activity: (1) return to the wait corner, defined as the start of the negative animal-corner velocity; (2) arrival to wait corner, when animal-corner distance <0.5 pix; (3) turn start, when angular velocity >0.01 degrees/s; (4) evaluation, the moment of lick port infrared sensor crossing. Spiking activity was convolved with a gaussian function (sigma = 25 ms) and rescaled through linear interpolation (*Kobak et al., 2016a*) to the average segment duration across all trials which allowed to construct peri-event time histograms (PETH) (*Figures 2, 3 and 5*).

For *Figure 2F, G and I* z-score was obtained for each unit across all conditions (eight stimuli, correct, and incorrect) and averaged according to the specified conditions. Statistical testing performed in *Figure 2I* was done with the Wilcoxon test.

## Linear regressions

Simple linear regressions were employed with some variables, for example, reaction time in *Figure 1E, H and J* and individual variables in the examples shown in *Figure 4A–C*. Significance testing was done through permutation (n=1000).

Multiple regression models were used in *Figure 4*. Data was obtained by generating a PETH with 0.2 s sliding windows (every 50 ms) and obtaining the value for the different variables tested (*Figure 4—figure supplement 1*) through interpolation. The models were fitted through 10-fold cross-validation using an L1 (LASSO) regularization term. Cross-validated $R^2$ values were obtained from untrained data for each fold and reported as the 10-fold mean. Given that regressors were not perfectly independent (*Figure 4—figure supplement 1*), the regularization term penalized the magnitude of the coefficients.

Partial correlation coefficient analyses (*Figure 4H–J*) were obtained by fitting a reduced model lacking the variable to be studied and using the residuals to obtain the correlation coefficient. Statistical significance was obtained by pseudo-random permutation (n=1000) of values. For distance to goal, the reduced model included all the kinematic variables (*Figure 4F*) and no spatio-temporal ones (given their high level of correlation, *Figure 4—figure supplement 1*). For angular velocity, the reduced model included all kinematic and spatio-termporal except the variable in question; for body speed, the reduced model also excluded head speed.

For *Figure 3—figure supplement 2*, we computed the loss of predictive power, delta $R^2$, for each of the four variables tested, similar to *Musall et al., 2019*. We created reduced models in which the

datapoints for the specified variable were shuffled. The difference in explained variance between the full and the reduced model equals the contribution (delta R2) of that variable to the model. This provides a metric for unique information (not shared by other variables) that each given variable contributes to the model.

## Population level analyses

Population level analyses were employed several times throughout the study (*Figures 3 and 5–7*), using standard Principal Component Analysis (PCA) and demixed Principal Component Analysis (dPCA) procedures (*Kobak et al., 2016a*).

Matrix construction varied according to the period and conditions used. For all population level analyses individual units recorded from all sessions and all animals were pooled to construct pseudo-simultaneous population response of data mostly recorded separately. For *Figure 3* condition averages were obtained for Go and evaluation periods, 0.3 s before turn start and 2.5 after lick port arrival. The dataset consisted of n=118 units recorded for 16 conditions (eight stimuli, with correct and incorrect responses) for which PETH were obtained as described before. We included units with at least two trials per condition. Note that responses can be classified by the direction of turn (Side), where for each stimulus correct and incorrect responses include turns to the right and left. Further, for left and right turns, there is stimulus gradation (0, +/-0.4, +/-0.6, +/-0.8), corresponding to the difficulty of the task. Thus, the 16 conditions per unit were used to construct a matrix of NxSxDxCxT, where N is the total of units (n=118), S is side (two conditions), D is difficulty (four conditions), C is correct and incorrect responses (two conditions), and T is time.

This matrix was used to calculate instantaneous variance (*Figure 3A*) for the different conditions with the following formulas:

$$VarSide_{INS}\left(t\right) = \frac{1}{N}\sum_{i=i}^{N}\sum_{S=1}^{2}\left(r^i\left(t,s\right) - r^i\left(t\right)\right)^2$$

$$VarCorrect_{INS}\left(t\right) = \frac{1}{N}\sum_{i=i}^{N}\sum_{c=1}^{2}\left(r^i\left(t,c\right) - r^i\left(t\right)\right)^2$$

$$VarDifficulty_{INS}\left(t\right) = \frac{1}{N}\sum_{i=i}^{N}\sum_{d=1}^{4}\left(r^i\left(t,d\right) - r^i\left(t\right)\right)^2$$

To compute the population cosine similarity presented in *Figure 3H–I* we used the aforementioned matrix, mean-subtracted it to remove condition-independent (temporal) dynamics. Then the following formula to all the population vector pairs at different timepoints:

$$CS = \frac{r^i_{t1} \cdot r^i_{t2}}{\left\|r^i_{t1}\right\| \left\|r^i_{t2}\right\|}$$

For PCA analysis, a covariance matrix was constructed by obtaining the pairwise covariance across all conditions. Eigenvalues and eigenvectors were then computed and sorted by variance. For *Figure 5—figure supplement 1* and *Figure 7K* the first three components were used to project the population responses. For *Figure 6I*, the fourth PC was used. Projections were obtained by computing the dot product with the eigenvector.

To assess the dimensionality of the population activity we compared the eigenvalues obtained from the data with those obtained with per-unit time-shuffled matrices. This destroys the time-dependent correlations and thus generates PCs due to chance. We performed this 1000 times and considered first PCs that explained more variance than the time-shuffled ones with a p<0.01 as statistically significant.

The details and mathematical procedure for dPCA analysis have been outlined elsewhere (*Kobak et al., 2016a*). Briefly, the method decomposes neural activity into the different chosen task variables to produce marginalized covariance matrices. The supervised part of the algorithm consists of choosing the variables to be analyzed. The unsupervised part of the algorithm uses a similar analysis to that of PCA on the marginalized covariance matrices. Our analyses were largely based on an available Python implementation (https://github.com/machenslab/dPCA, *Kobak et al., 2016b*). In

*Figure 4* we marginalized the population activity on time (condition-independent dynamics), Side, Difficulty, and Correct vs incorrect.

The dPCA analysis presented in *Figure 5* was performed similarly to that of *Figure 3* but using different time periods and conditions. For each session, model trajectories for Go, left and right, were constructed by averaging the body speed and angular velocity. Then, we computed the correlation between the model trajectory and individual trial trajectories, both from the Return and Go periods. We selected those with a correlation coefficient of ≥0.75. Finally, we obtained PETH for the selected, highly similar trajectories.

For each unit, we obtained four conditions: Return left and right, and Go left and right. With these we constructed a matrix NxCxS, where N is units (n=118), C is context (two conditions, Return vs Go) and S is side (two conditions, left and right). Thus, the marginalizations were performed on the basis of these two variables and time (condition-independent dynamics), mainly to extract the contextual (Return vs Go) population dynamics.

## Spike - cyclic behavior phase analysis

We analyzed how spiking was coupled to two cyclic behaviors, gait and licking. For gait, we computed the paw position by projecting the position onto the longitudinal axis of the mouse. We found individual gait cycles by finding the peaks of this signal (*Figure 6A*). We used movement periods that had at least four cycles/paces. We applied a band-passed filter (3–6 Hz) to the signal and extracted the instantaneous phase through a Hilbert transform. We then computed the phase at each spike timestamp in a 400 ms window around the peak (which includes around two gait cycles) and generated a spike- gait-phase histogram. This was done with the hindpaw contralateral to recorded hemisphere since hindpaws display a higher amplitude than forepaws. The average vector was computed and tested for directional bias through the Rayleigh test for each of the eight strides analyzed (*Figure 6E*). Mean phase and magnitude were computed by averaging the individual per-unit x and y-axis components.

For licking behavior, we computed the lick probability around an individual lick and applied a 4–10 Hz band-pass filter. This signal was then used to compute the spike - lick-phase in a 400 ms window (around two licks) like that performed with gait.

## Support vector classifiers (SVCs)

To understand whether the population could store information for several variables simultaneously we trained SVCs using the implementation in scikit (*Pedregosa et al., 2011*). For gait cycle, the same matrix used for population dimensionality reduction through PCA was used. Each time vector, containing n=118 unit averages was labeled according to the variables analyzed: context, step number, paw position, and paw velocity. Note that continuous signals (position and velocity) were downsampled to five states to allow for 10-fold cross-validation. To assess for statistical significance, we repeated the procedure on data with shuffled labels (n=1000) and assessed the probability of a higher accuracy of the original model. Given that we could not decode paw velocity from the original matrix, we computed the PC4 weighted matrix by using the eigenvector derived from PCA (*Figure 6F*).

For licking we employed the same matrix used for PCA (part of which is shown in *Figure 7J*) and trained models for lick phase, left-right licks, lick number, and correct vs incorrect licks. To assess significance, a similar permutation procedure was performed.

## Acknowledgements

VRAK is a doctoral student from the Programa de Doctorado en Ciencias Biomédicas, Universidad Nacional Autónoma de México (UNAM) and has received CONAHCyT fellowship CVU 808903. This work was supported by CONACyT grant 220412, Fronteras de la Ciencia CONACyT grants 2022, 2019/154039, CF-2023-I-305, the DGAPA-PAPIIT-UNAM grants IN226517, IN203420, IN203123, and the Moshinsky fellowship to FT. The authors would like to thank A Cesar Poot-Hernández and Carlos A Peralta-Alvaréz from the Bioinformatics Unit for assistance, Gabriel Diaz-deLeon for proofreading the manuscript, and Pavel Rueda-Orozco for helpful discussions.

# Additional information

## Funding

| Funder | Grant reference number | Author |
|---|---|---|
| Consejo Nacional de Humanidades, Ciencias y Tecnologías | 808903 | Anil K Verma Rodriguez |
| Consejo Nacional de Humanidades, Ciencias y Tecnologías | CF-2023-I-305 | Fatuel Tecuapetla |
| Dirección General de Asuntos del Personal Académico, Universidad Nacional Autónoma de México | IN226517 | Fatuel Tecuapetla |
| Fundación Marcos Moshinsky | Catedra 2019 | Fatuel Tecuapetla |
| Consejo Nacional de Humanidades, Ciencias y Tecnologías | 2022 | Fatuel Tecuapetla |
| Dirección General de Asuntos del Personal Académico, Universidad Nacional Autónoma de México | IN203420 | Fatuel Tecuapetla |
| Dirección General de Asuntos del Personal Académico, Universidad Nacional Autónoma de México | IN203123 | Fatuel Tecuapetla |
| Consejo Nacional de Humanidades, Ciencias y Tecnologías | 2019/154039 | Fatuel Tecuapetla |
| Consejo Nacional de Humanidades, Ciencias y Tecnologías | 220412 | Fatuel Tecuapetla |

The funders had no role in study design, data collection and interpretation, or the decision to submit the work for publication.

## Author contributions

Anil K Verma Rodriguez, Conceptualization, Data curation, Formal analysis, Investigation, Visualization, Methodology, Writing – original draft, Project administration, Writing – review and editing; Josue O Ramírez-Jarquin, Resources, Help with animals and resources adquisition; Román Rossi-Pool, Supervision; Fatuel Tecuapetla, Conceptualization, Resources, Supervision, Funding acquisition, Investigation, Writing – original draft, Project administration, Writing – review and editing

## Author ORCIDs

Anil K Verma Rodriguez ![ORCID] https://orcid.org/0009-0008-5374-4958
Fatuel Tecuapetla ![ORCID] https://orcid.org/0000-0002-9191-6553

## Ethics

All procedures were approved by the Institutional Committee for the Care and Use of Laboratory Animals at the Instituto de Fisiología Celular (Protocol number FTA-121-17). Universidad Nacional Autónoma de México. and the National Norm for the Useof Animals (NOM-062-ZOO-1999).

Reviewer #1 (Public review): https://doi.org/10.7554/eLife.98159.3.sa1
Reviewer #2 (Public review): https://doi.org/10.7554/eLife.98159.3.sa2
Author response https://doi.org/10.7554/eLife.98159.3.sa3

## Additional files

### Supplementary files

Supplementary file 1. Summary statistical table. Details of the statistics used for each figure.

MDAR checklist

### Data availability

Data publicly available: All timestamps data of the current study, instructions and an example code have been made publicly available in Zenodo. Further information and requests for data and code should be directed to and will be fulfilled by the lead contact, Fatuel Tecuapetla.

The following dataset was generated:

| Author(s) | Year | Dataset title | Dataset URL | Database and Identifier |
|---|---|---|---|---|
| Verma-Rodríguez AK, Tecuapetla F, Ramírez-Jarquín JO, Rossi-Pool R | 2025 | Basal ganglia output coding - entopeduncular nucleus - of contextual kinematics and reward in the freely moving mouse | https://doi.org/10.5281/zenodo.14846394 | Zenodo, 10.5281/zenodo.14846394 |

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
