## [Editor Report · eLife Assessment]

This **valuable** study reports on electrophysiological recording of the spiking activity of single neurons in the entopeduncular nucleus (EPN) in freely-moving mice performing an auditory discrimination task. The data show that the activity of single EPN neurons is modulated by reward and movement kinematics, with the latter further affected by task contexts (e.g. movement toward or away from a reward location). The results provide **solid** evidence for the conclusions. There is some ambiguity as to whether the data contain the population of EPN neurons characterized in previous studies that obtained different results. Investigations separating confounding factors would be of benefit. Nonetheless, the work is overall of interest to those who study how the basal ganglia, particularly the EPN, contribute to behavior.

---

## [Referee Report · Reviewer #1 (Public review)]

The authors in this paper investigate the nature of the activity in the rodent EPN during a simple freely moving cue-reward association task. Given that primate literature suggest movement coding whereas other primate and rodent studies suggest mainly reward outcome coding in the EPNs, it is important try to tease apart the two views. Through careful analysis of behavior kinematics, position, and the neural activity in the EPNs, the authors reveal an interesting and complex relationship between the EPN and mouse behavior.

Strengths:

(1) The authors use a novel freely moving task to study EPN activity, which displays rich movement trajectories and kinematics. Given that previous studies have mostly looked at reward coding during head fixed behavior, this study adds a valuable dataset to the literature.

(2) The neural analysis is rich and thorough. Both single neuron level and population level (i.e. PCA) analysis are employed to reveal what EPN encodes.

Discussion:

EPN is one of the major output nuclei of the basal ganglia. What information is present within EPN is still unclear, and under investigation. The authors have used electrophysiology to determine the nature of information present within EPN that is likely to be valuable to the field. Future studies should try to address whether this information is specific to certain cell types within EPN or whether there is topography within EPN that reflects the kinematic information present within EPN. This will require more careful dissection of EPN activity based on anatomy. Future experiments should also consider tasks that isolate a single limb (i.e. joystick tasks) in order to better understand the kinematic encoding of forelimb movement. This, combined with recording in forelimb encoding region of EPN, should give us insights into the nature of kinematic control of EPN. Overall, this study will be useful to inspire future investigations in the function of EPN.

---

## [Referee Report · Reviewer #2 (Public review)]

This paper examined how the activity of neurons in the entopeduncular nucleus (EPN) of mice relates to kinematics, value, and reward. The authors recorded neural activity during an auditory cued two-alternative choice task, allowing them to examine how neuronal firing relates to specific movements like licking or paw movements, as well as how contextual factors like task stage or proximity to a goal influence the coding of kinematic and spatiotemporal features. The data shows that the firing of individual neurons is linked to kinematic features such as lick or step cycles. However, the majority of neurons exhibited activity related to both movement types, suggesting that EPN neuronal activity does not merely reflect muscle-level representations. This contradicts what would be expected from traditional action selection or action specification models of the basal ganglia.

The authors also show that spatiotemporal variables account for more variability compared to kinematic features alone. Using demixed Principal Component Analysis, they reveal that at the population level, the three principal components explaining the most variance were related to specific temporal or spatial features of the task, such as ramping activity as mice approached reward ports, rather than trial outcome or specific actions. Notably, this activity was present in neurons whose firing was also modulated by kinematic features, demonstrating that individual EPN neurons integrate multiple features. A weakness is that what the spatiotemporal activity reflects is not well specified. The authors suggest some may relate to action value due to greater modulation when approaching a reward port, but acknowledge action value is not well parametrized or separated from variables like reward expectation.

A key goal was to determine whether activity related to expected value and reward delivery arose from a distinct population of EPN neurons or was also present in neurons modulated by kinematic and spatiotemporal features. In contrast to previous studies (Hong & Hikosaka 2008 and Stephenson-Jones et al., 2016), the current data reveals that individual neurons can exhibit modulation by both reward and kinematic parameters. Two potential differences may explain this discrepancy: First, the previous studies used head-fixed recordings, where it may have been easier to isolate movement versus reward-related responses. Second, those studies observed prominent phasic responses to the delivery or omission of expected rewards - responses that are present but not common in the current paper. This suggests a possibility that the VGlut2+ EPN neurons that project to the LHb were under/not sampled, antidromic or optogenetic tagging would have been needed to confirm the identity of the populations that were recorded. Alternatively, in the head-fixed recordings, kinematic/spatial coding may have gone undetected due to the forced immobility.

Overall, this paper offers needed insight into how the basal ganglia output encodes behavior. The EPN recordings from freely moving mice clearly demonstrate that individual neurons integrate reward, kinematic, and spatiotemporal features, challenging traditional models. However, the specific relationship between the spatiotemporal activity and factors like action value remains unclear.

---

## [Author Response]

The following is the authors’ response to the original reviews.

**Reviewer #1 (Public Review):**
The authors in this paper investigate the nature of the activity in the rodent EPN during a simple freely moving cue-reward association task. Given that primate literature suggests movement coding whereas other primate and rodent studies suggest mainly reward outcome coding in the EPNs, it is important to try to tease apart the two views. Through careful analysis of behavior kinematics, position, and neural activity in the EPNs, the authors reveal an interesting and complex relationship between the EPN and mouse behavior.Strengths:(1) The authors use a novel freely moving task to study EPN activity, which displays rich movement trajectories and kinematics. Given that previous studies have mostly looked at reward coding during head-fixed behavior, this study adds a valuable dataset to the literature. (2) The neural analysis is rich and thorough. Both single neuron level and population level (i.e. PCA) analysis are employed to reveal what EPN encodes.

Thank you very much for this appreciation.

Weaknesses:(1) One major weakness in this paper is the way the authors define the EPN neurons. Without a clear method of delineating EPN vs other surrounding regions, it is not convincing enough to call these neurons EPNs solely from looking at the electrode cannula track from Figure 2B. Indeed, EPN is a very small nucleus and previous studies like Stephenson-Jones et al (2016) have used opto-tagging of Vglut2 neurons to precisely label EPN single neurons. Wallace et al (2017) have also shown the existence of SOM and PV-positive neurons in the EPN. By not using transgenic lines and cell-type specific approaches to label these EPN neurons, the authors miss the opportunity to claim that the neurons recorded in this study do indeed come from EPN. The authors should at least consider showing an analysis of neurons slightly above or below EPN and show that these neurons display different waveforms or firing patterns.

We thank the reviewer for their comment, and we thank the opportunity to expand on the inclusion criteria of studied units after providing an explanation.

As part of another study, we performed experiments recording in EPN with optrodes and photoidentification in PV-Cre animals. We found optoidentified units in both: animals with correct placement (within the EPN) and on those with off-target placement (within the thalamus or medial to the EPN). Thus, despite the use of Cre animals, we relied on histology to ensure correct EPN recording. We believe that the optotagging based purely on neural makers such as PV, SOM, VGLUT, VGAT would not provide a better anatomical delineation of the EPN since adjacent structures are rich in those same markers. The thalamic reticular nucleus is just dorsal to the EPN and it has been shown to express both SOM and PV (Martinez-Garcia et al., 2020).

On the other hand, the lateral hypothalamus (just medial to the EPN) also expresses vGlut2 and SOM. Stephenson-Jones (2016), Extended Data Figure 1, panel g, shows vGluT2 and somatostatin labeling of neurons, with important expression of neurons dorsal, ventral and medial to the EPN. Thus, we believe that viral strategies relying on single neuronal markers still depend on careful histological analysis of recording sites.

A combination of neural markers or more complex viral strategies might be more suitable to delineate the EPN. As an example, for anatomical tracing Stephenson-Jones et al. 2016 performed a rabies-virus based approach involving retrogradely transported virus making use of projection sites through two injections. Two step viral approaches were also performed in Wallace, M. et al. 2017. We attempted to perform a two-step viral approach, using an anterogradely transported Cre-expressing virus (AAV1.hSyn.Cre.WPRE.hGH) injected into the striatum and a second Cre dependent ChR2 into the EPN. However, our preliminary experiments showed that this double viral approach had a stark effect decreasing the performance of animals during the task (we attempted re-training 2-3 weeks after viral infections and animals failed to turn to the contralateral side of the injections). We believe that this approach might have had a toxic effect (Zingg et al., 2017).

To this point, a recent paper (Lazaridis et al., 2019) repeated an optogenetic experiment performed in the Stephenson-Jones et al. study, using a set of different viral approaches and concluded that increasing the activity of GPi-LHb is not aversive, as it had been previously reported. Thus, future studies attempting to increase anatomical specificity are a must, but they will require using viral approaches amenable to the behavioral paradigm.

We attempted to find properties regarding waveforms, firing rate, and firing patterns from units above or below, however, we did not find a marker that could generate a clear demarcation. We show here a figure that includes the included units in this study as well as excluded ones to show that there is a clear overlap.

Finally, we completely agree with the reviewer in that there is still room for improvement. We have further expanded the Methods section to explain better our efforts to include units recorded within the EPN. Further, we have added a paragraph within the Discussion section to point out this limitation.

Methods:

“Recordings. Movable microwire bundles (16 microwires, 32 micrometers in diameter, held inside a cannula, Innovative Neurophysiology, Durham, NC) were stereotaxtically implanted just above the entopeduncular nucleus (-0.8 AP, 1.7 ML, 3.9 DV). Post surgical care included antibiotic, analgesic and antiinflammatory pharmacological treatment. After 5 days of recovery, animals were retrained for 1-2 weeks. Unitary activity was recorded for 2-6 days at each dorsoventral electrode position and the session with the best electrophysiological (signal to noise ratio (>2), stability across time) and behavioral [performance, number of trials (>220)] quality was selected. Microwire electrodes were advanced in 50 micrometer dorsoventral steps for 500 micrometers in total. After experiment completion, animals were perfused with a 4% paraformaldehyde solution. Brains were extracted, dehydrated with a 30% sucrose solution and sectioned in a cryostat into 30micron thick slices. Slices were mounted and photographed using a light microscope. Microwire tracks of the 16-microwire bundle were analyzed (Fig. 2A-B) and only animals with tracks traversing the EPN were selected (6 out of 10). Finally, we located the final position of microwire tips and inferred the dorsoventral recording position of each of the recording sessions. Only units recorded within the EPN were included.”

Discussion:

“A weakness of the current study is the lack of characterization of neuronal subtypes. An area of opportunity for future research could be to perform photo-identification of neuronal subtypes within the EPN which could contribute to the overall description of the information representation. Further, detailed anatomical viral vector strategies could aid to improve anatomical localization of recordings, reduce reliance on histological examination, and solve some current controversies (Lazaridis et al., 2019).”

(2) The authors fail to replicate the main finding about EPN neurons which is that they encode outcome in a negative manner. Both Stephenson-Jones et al (2016) and Hong and Hikosaka (2008) show a reward response during the outcome period where firing goes down during reward and up during neutral or aversive outcome. However, Figure 2 G top panel shows that the mean population is higher during correct trials and lower during incorrect trials. This could be interesting given that the authors might try recording from another part of EPN that has not been studied before. However, without convincing evidence that the neurons recorded are from EPN in the first place (point 1), it is hard to interpret these results and reconcile them with previous studies.

We really thank the reviewer for pointing out that we need to better explain how EPN units encode outcome. We now provide an additional panel in Figure 4, its corresponding text in the results section and a new paragraph in the discussion related to this comment.

We believe that we do indeed recapitulate findings of both of Stephenson-Jones et al (2016) and Hong and Hikosaka (2008). Both studies focus on a specific subpopulation of GPi/EPN neurons that project to the lateral habenula (LHb). Stephenson-Jones et al (2016) posit that GPi-LHb neurons (which they opto-tag as vGluT2) exhibit a decreased firing rate during rewarding outcomes. Hong and Hikosaka (2008) antidromically identified LHb projecting neurons through within the GPi and found reward positive and reward negative neurons, which were respectively modulated either by increasing or decreasing their firing rate with a rewarding outcome (red and green dots on the x-axis of Figure 5A in their paper).

As the reviewer pointed out the zScore may be misleading. Therefore, in our study we also decomposed population activity on reward axis through dPCA. When marginalizing for reward in Figure 3F, we find that the weights of individual units on this axis are centered around zero, with positive and negative values (Figure 3F, right panel). Thus, units can code a rewarding outcome as either an increase or a decrease of activity. We show example units of such modulation in Figure 3-1g and h.

We had segregated our analysis of spatio-temporal and kinematic coding upon the reward coding of units in Figure 4L-M. Yet, following this comment and in an effort of further clarifying this segregation, we introduced panels with the mean zScore of units during outcome evaluation in Figure 4L.

We amended the main text to better explain these findings.

“Previous reports suggest that EPN units that project to the lateral habenula encode reward as a decrease in firing rate. Thus, we wished to ask whether reward encoding units can code kinematic and spatio-temporal variables as well.

To this end, we first segregated units upon their reward coding properties: reward positive (which increased activity with reward) and reward negative units (which decreased activity with reward). We performed auROC on the 250ms after head entry comparing rewarded trials and incorrect trails (p<0.001, permutation test). Mean activity of reward insensitive, positive and negative units is shown in Fig. 4L. Next, we performed a dimensionality reduction on the coefficients of the model that best explained both contexts (kinematic + spatio-temporal model on pooled data) using UMAP (McInnes et al., 2018). We observe a continuum rather than discrete clusters (Fig. 4L). Note that individual units are color coded according to their responsivity to reward. We did not find a clear clustering either.”

Paragraph added in the discussion:

“In this study, we found that rewarding outcomes can be represented by EPN units through either an increase or a decrease in firing rate (Fig. 3F, 3-1g-h, 4L). While Stephenson-Jones et al., 2016 found that lateral habenula (LHb)-projecting neurons within the EPN of mice primarily encoded rewarding outcomes by a decrease in firing rate, Hong and Hikosaka, 2008 observed that in primates, LHb-projecting units could encode reward through either a decrease or an increase in firing rate. Thus, our results align more closely with the latter study, which also employed an operant conditioning task.”

(3) The authors say that: 'reward and kinematic doing are not mutually exclusive, challenging the notion of distinct pathways and movement processing'. However, it is not clear whether the data presented in this work supports this statement. First, the authors have not attempted to record from the entire EPN. Thus it is possible that the coding might be more segregated in other parts of EPN. Second, EPNs have previously been shown to display positive firing for negative outcomes and vice versa, something which the authors do not find here. It is possible that those neurons might not encode kinematic and movement variables. Thus, the authors should point out in the main text the possibility that the EPN activity recorded might be missing some parts of the whole EPN.

We thank the reviewer for the opportunity to expand on this topic. We believe it is certainly possible that other not-recorded regions of the EPN might exhibit greater segregation of reward and kinematics. However, we considered it worthwhile pointing out that from the dataset collected in this study reward-sensitive units encode kinematics in a similar fashion to reward-insensitive ones (Fig. 4L,M). Moreover, we asked specifically whether reward-negative units (that decrease firing rate with rewarding outcomes, as previously reported) could encode kinematics and spatio-temporal variables with different strength than reward-insensitive ones and could not find significant differences (Fig. 4M).

We did indeed find units that displayed decreased firing rate upon rewarding outcomes, as has been previously reported. We have addressed this fact more thoroughly in point (2).

Finally, we agree with the reviewer that the dataset collected in this study is by no means exhaustive of the entire EPN and have thus included a sentence pointing this out in the Discussion section:

“Given that we did not record from the entire EPN, it is still possible that another region of the nucleus might exhibit more segregation.”

(4) The authors use an IR beam system to record licks and make a strong claim about the nature of lick encoding in the EPN. However, the authors should note that IR beam system is not the most accurate way of detecting licks given that any object blocking the path (paw or jaw-dropping) will be detected as lick events. Capacitance based, closed-loop detection, or video capturing is better suited to detect individual licks. Given that the authors are interested in kinematics of licking, this is important. The authors should either point this out in the main text or verify in the system if the IR beam is correctly detecting licks using a combination of those methods.

We thank the reviewer for the opportunity of clarifying the lick event acquisition. We have experience using electrical alternatives to lickometers; however, we believe they were not best suited to this application. Closed-loop lickometers generally use a metallic grid upon which animals stand so that the loop can be closed; however, we wanted to have a transparent floor. We have found capacitance based lickometers to be useful in head-fixed conditions but have noticed that they are very dependent on animal position and proximity of other bodyparts such as limbs. Given the freely moving aspect of the task this was difficult to control. Finally, both electric alternatives for lickometers are more prone to noise and may introduce electrical artifacts that might contaminate the spiking signal. This is why we opted to use a slit in combination with an IR beam that would only fit the tongue and that forced enough protrusion such that individual licks could be monitored. Further, the slit could not fit other body-parts like the paw or jaw. We have now included a video (Supp. Video 2) showing a closeup of this behavior that better conveys how the jaw and paw do not fit inside the slit. The following text has been added in the corresponding methods section:

“The lickometer slit was just wide enough to fit the tongue and deep enough to evoke a clear tongue protrusion.”

**Reviewer #1 (Recommendations For The Authors):**
(1)The authors should verify using opto-tagging of either Vglut2, SOM, or PV neurons whether they can see the same firing pattern. If not, the authors should address this weakness in the paper.

We thank the reviewer for this important point, we have provided a more detailed reply above.

(2)The way dPCA or PCA is applied to the data is not stated at all in the main text. Are all units from different mice combined? Or applied separately for each mouse? How does that affect the interpretation of the data? At least a brief text should be included in the main text to guide the readers.

We thank the reviewer for pointing out this important omission. We have included an explanation in the Methods section and in the Main text.

Methods:

“For all population level analyses individual units recorded from all sessions and all animals were pooled to construct pseudo-simultaneous population response of combined data mostly recorded separately.”

Main text:

“For population level analyses throughout the study, we pooled recorded units from all animals to construct a pseudo-simultaneous population.”

Discussion:

“…(from pooled units from all animals to construct a pseudo-simultaneous population, which assumes homogeneity across subjects)”

(3) The authors argue that they do not find 'value coding' in this study. However, the authors never manipulate reward size or probability, but only the uncertainty or difficulty of the task. This might be better termed 'difficulty', and it is difficult to say whether this correlates with value in this task. For instance, mice might be very confident about the choice, even for an intermediate frequency sweep, if the mouse had waited long enough to hear the full sweep. In that case, the difficulty would not correlate with value, given that the mouse will think the value of the port it is going to is high. Thus, authors should avoid using the term value.

We agree with the reviewer. We have modified the text to specify that difficulty was the variable being studied and added the following sentence in the Discussion:

“It is still possible that by modifying reward contingencies such as droplet size value coding could be evidenced.”

(4) How have the authors obtained Figure 7D bottom panel? It is unclear at all what this correlation represents. Are the authors looking at a correlation between instantaneous firing rate and lick rate during a lick bout?

We thank the reviewer for pointing out that omission. It is indeed correlation coefficient between the instantaneous firing rate and the instantaneous lick rate for a lick bout. We have included labeling in Figure 7D and pointed this out in the main text:

“Fig.7D, lower panel shows the correlation coefficient between the instantaneous firing rate and the instantaneous lick rate within a lick bout for all units.”

**Reviewer #2 (Public Review):**
This paper examined how the activity of neurons in the entopeduncular nucleus (EPN) of mice relates to kinematics, value, and reward. The authors recorded neural activity during an auditory-cued two-alternative choice task, allowing them to examine how neuronal firing relates to specific movements like licking or paw movements, as well as how contextual factors like task stage or proximity to a goal influence the coding of kinematic and spatiotemporal features. The data shows that the firing of individual neurons is linked to kinematic features such as lick or step cycles. However, the majority of neurons exhibited activity related to both movement types, suggesting that EPN neuronal activity does not merely reflect muscle-level representations. This contradicts what would be expected from traditional action selection or action specification models of the basal ganglia.The authors also show that spatiotemporal variables account for more variability compared to kinematic features alone. Using demixed Principal Component Analysis, they reveal that at the population level, the three principal components explaining the most variance were related to specific temporal or spatial features of the task, such as ramping activity as mice approached reward ports, rather than trial outcome or specific actions. Notably, this activity was present in neurons whose firing was also modulated by kinematic features, demonstrating that individual EPN neurons integrate multiple features. A weakness is that what the spatiotemporal activity reflects is not well specified. The authors suggest some may relate to action value due to greater modulation when approaching a reward port, but acknowledge action value is not well parametrized or separated from variables like reward expectation.

We thank the reviewer for the comment. We indeed believe that further exploring these spatiotemporal signals is important and will be the subject of future studies.

A key goal was to determine whether activity related to expected value and reward delivery arose from a distinct population of EPN neurons or was also present in neurons modulated by kinematic and spatiotemporal features. In contrast to previous studies (Hong & Hikosaka 2008 and Stephenson-Jones et al., 2016), the current data reveals that individual neurons can exhibit modulation by both reward and kinematic parameters. Two potential differences may explain this discrepancy: First, the previous studies used head-fixed recordings, where it may have been easier to isolate movement versus reward-related responses. Second, those studies observed prominent phasic responses to the delivery or omission of expected rewards - responses largely absent in the current paper. This absence suggests a possibility that neurons exhibiting such phasic "reward" responses were not sampled, which is plausible since in both primates and rodents, these neurons tend to be located in restricted topographic regions. Alternatively, in the head-fixed recordings, kinematic/spatial coding may have gone undetected due to the forced immobility.

Thank you for raising this point. Nevertheless, there is some phasic activity associated with reward responses, which can be seen in the new panel in Figure 4L.

Overall, this paper offers needed insight into how the basal ganglia output encodes behavior. The EPN recordings from freely moving mice clearly demonstrate that individual neurons integrate reward, kinematic, and spatiotemporal features, challenging traditional models. However, the specific relationship between spatiotemporal activity and factors like action value remains unclear.

We really appreciate this reviewer for their valuable comments.

**Reviewer #2 (Recommendations For The Authors):**
One small suggestion is to make sure that all the panels in the figures are well annotated. I struggled in places to know what certain alignments or groupings meant because they were not labelled. An example would be what do the lines correspond to in the lower panels of Figure 2D and E. I could figure it out from other panels but it would have helped if each panel had better labelling.

Thanks for pointing this out, we have improved labelling across the figures and corrected the specific example you have pointed out.

The paper is very nice though. Congratulations!

Thank you very much.

**Editor's note:**
Should you choose to revise your manuscript, please include full statistical reporting including exact p-values wherever possible alongside the summary statistics (test statistic and df) and 95% confidence intervals. These should be reported for all key questions and not only when the p-value is less than 0.05 in the main manuscript.

We thank the editor for the comment. A statistics table has been added.

References:

Lazaridis, I., Tzortzi, O., Weglage, M., Märtin, A., Xuan, Y., Parent, M., Johansson, Y., Fuzik, J., Fürth, D., Fenno, L. E., Ramakrishnan, C., Silberberg, G., Deisseroth, K., Carlén, M., & Meletis, K. (2019). A hypothalamus-habenula circuit controls aversion. Molecular Psychiatry, 24(9), 1351–1368. https://doi.org/10.1038/s41380-019-0369-5

Martinez-Garcia, R. I., Voelcker, B., Zaltsman, J. B., Patrick, S. L., Stevens, T. R., Connors, B. W., & Cruikshank, S. J. (2020). Two dynamically distinct circuits drive inhibition in the sensory thalamus. Nature, 583(7818), 813–818. https://doi.org/10.1038/s41586-0202512-5

McInnes, L., Healy, J., Saul, N., & Großberger, L. (2018). UMAP: Uniform Manifold Approximation and Projection. Journal of Open Source Software, 3(29), 861. https://doi.org/10.21105/joss.00861

Zingg, B., Chou, X. lin, Zhang, Z. gang, Mesik, L., Liang, F., Tao, H. W., & Zhang, L. I. (2017). AAV-Mediated Anterograde Transsynaptic Tagging: Mapping Corticocollicular Input-Defined Neural Pathways for Defense Behaviors. Neuron, 93(1), 33–47. https://doi.org/10.1016/j.neuron.2016.11.045